# Dataset Inference for Self-Supervised Models

**Adam Dziedzic**[*][†]**, Haonan Duan**[†]**, Muhammad Ahmad Kaleem**[†]**, Nikita Dhawan,**
**Jonas Guan, Yannis Cattan, Franziska Boenisch, Nicolas Papernot**
University of Toronto and Vector Institute

## Abstract

Self-supervised models are increasingly prevalent in machine learning (ML) since they reduce the need for expensively labeled data. Because of their versatility in downstream applications, they are increasingly used as a service exposed via public APIs. At the same time, these encoder models are particularly vulnerable to model stealing attacks due to the high dimensionality of vector representations they output. Yet, encoders remain undefended: existing mitigation strategies for stealing attacks focus on supervised learning. We introduce a new dataset inference defense, which uses the private training set of the victim encoder model to attribute its ownership in the event of stealing. The intuition is that the log-likelihood of an encoder's output representations is higher on the victim's training data than on test data if it is stolen from the victim, but not if it is independently trained. We compute this log-likelihood using density estimation models. As part of our evaluation, we also propose measuring the fidelity of stolen encoders and quantifying the effectiveness of the theft detection without involving downstream tasks; instead, we leverage mutual information and distance measurements. Our extensive empirical results in the vision domain demonstrate that dataset inference is a promising direction for defending self-supervised models against model stealing.

## 1 Introduction

The self-supervised learning (SSL) paradigm enables pre-training models with unlabeled data to learn generally useful domain knowledge and then transfer the knowledge to solve specific downstream tasks. The ability to learn from unlabeled data alleviates the high costs of labeling large datasets [24], and the transfer learning setup reduces the computational costs of retraining. These advantages have made SSL increasingly popular [20] in domains like vision [5], language [12], and bioinformatics [29].

Recently, commercial service providers like Cohere [1] and OpenAI [2] began offering paid query access to trained SSL encoders over public APIs. This exposes the encoders to black-box extraction attacks, *i.e.,* model stealing. In a model stealing attack, an attacker aims to train an approximate copy of a victim model by submitting carefully chosen queries and observing the victim's outputs. The high costs of data collection, preprocessing and model training make encoders valuable targets for stealing. For example, the training data of CLIP includes 400 million image and text pairs [37], while computation costs of training a large language model can exceed one million USD [41]. The threat of model stealing in SSL is real: researchers have demonstrated that encoders can be stolen at a fraction of the victim's training cost [14, 39]. Yet, most current defenses are designed for supervised models [15, 23, 35] and cannot be directly applied to encoders [14].

Dataset inference [31] is a state-of-the-art defense against model stealing in the supervised learning setting. The defense provides ownership resolution: it enables the model owner to make a strong statistical claim that a given model is a stolen copy of their own model by showing that this model is derivative of their own private training data. Dataset inference does not require retraining or overfitting the model to any form of explicit watermark [4] and has been shown to resist attacks from

---

[*]Corresponding and leading author: adam.dziedzic@utoronto.ca
[†]Equal contribution.

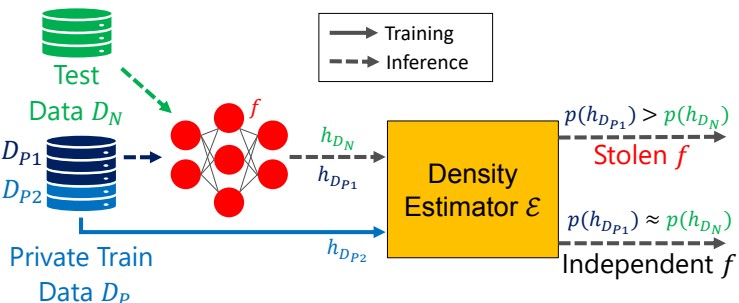

Figure 1: **Ownership Resolution for Encoders.** First, an arbitrator trains density estimator $\mathcal{E}$: divide $D_P$ into non-overlapping partitions $D_{P1}$ and $D_{P2}$, and train density estimator $\mathcal{E}$ using the representations $h_{D_{P2}}$ of $f$ on $D_{P2}$. Next, the arbitrator performs dataset inference: apply $\mathcal{E}$ on the representations of $D_{P1}$ and $D_N$ of the encoder $f$. For a stolen encoder, the log-likelihood of the representations $h_{D_{P1}}$ is significantly higher than $h_{D_N}$, while, for an independent encoder, the log-likelihoods of the representations are not significantly different.

adaptive adversaries [31]. These properties make the defense particularly attractive for SSL, as large encoders can be expensive to retrain, and performance is paramount because it carries over to all downstream tasks. However, the original dataset inference algorithm from [31] cannot be applied to encoders, because it relies on computing distances between data points and decision boundaries. These decision boundaries do not exist in SSL encoders since they are trained on unlabeled data.

We introduce a new dataset inference method (Figure 1) to defend against model stealing for encoders. Our algorithm is suitable for the high-dimensional outputs of SSL encoders and does not rely on labeled data or decision boundaries. Instead, it relies solely on the private training data of the victim encoder as a signature. Moreover, our algorithm retains the advantages of dataset inference for supervised models [31], namely, it does not require retraining or overfitting the SSL encoder.

Our key intuition is to identify stolen encoders by characterizing differences between an encoder's representations on its training data vs on unseen test data. The victim encoder and its derivatives, such as stolen copies, exhibit different behavior on the victim's private training data than on test data; independently trained encoders do not. These differences exist because encoders overfit to training data [18, 30]. Although for well-trained encoders the effect is minimal on any given data point, we show that when aggregated over many training points it provides a statistically strong signal. To identify the differences, we train a Gaussian Mixture Model (GMM), as an efficient general approximation, to model the distribution of an encoder's data representations from its training domain. We then use the GMMs to predict the log-likelihood of the encoders' representations of the victim's training set and a test set; derivatives of the victim encoder will have a higher log-likelihood on the training set than on the test set. We perform experiments on five datasets from the vision domain and show that we are able to distinguish between stolen and independent encoders even in cases when adversaries obfuscate the representations from the stolen encoders to hide the theft (*e.g.,* by shuffling the elements in the representation vectors or applying to them some form of a linear transformation).

As part of our evaluation, we also introduce new metrics to measure the fidelity of stolen encoders without involving downstream tasks and to quantify the effectiveness of theft detection. We compute scores directly on the representations using tools from information theory and distance metrics. These methods work well because losses used for stealing encoders directly minimize distances between representations of victim and stolen encoders. Our mutual information score to assess the quality of the stolen encoders is robust against obfuscations that an adversary might apply to the representations returned by a stolen encoder. Without any obfuscation, our cosine similarity score shows a clearer distinction between stolen and independent encoders. Finally, using these metrics, we observe that the higher the quality of the stolen encoders, the more confident our dataset inference defense becomes.

Our main contributions are as follows:

- We propose a new defense against model stealing attacks on encoders, by combining dataset inference with density estimation models for ownership resolution on unlabeled data.

- We are the first to design new metrics that quantify the quality of stolen encoders, which are derived from the mutual information and distances between representations.

- We evaluate our defense using five datasets from the computer vision domain and show that our defense can successfully identify stolen encoders with a strong statistical significance.

## 2 Related Work

In model stealing, an adversary queries the victim model, obtains outputs, and uses them to recreate a copy of the victim [42]. This is most commonly performed with black-box access, *e.g.,* via a public API. When stealing encoders in SSL, the goal of an adversary is to extract high-quality embeddings either to train a stolen copy that achieves high performance on downstream tasks, or to obtain faithful replicas of the victim's embeddings on the same inputs. Stolen encoders might be further used for model reselling, backdoor attacks, or membership inference [9].

While most past research on model stealing and defenses focuses on classifiers trained via supervised learning, recent work constructed new attacks that target encoders [14, 39]. The main differences between the attacks in these settings are that the outputs of encoders leak more information due to their higher dimensionalities [39], and the attacks require different loss functions. Inspired by contrastive learning, *Cont-Steal* [39] provides a method of stealing encoders using a loss function based on InfoNCE [5]. SSL extraction [14]—a general Siamese-network-based framework for stealing encoders—leverages losses including mean squared error, InfoNCE, Soft Nearest Neighbor, and Wasserstein distance. The authors empirically show that an adversary can steal an ImageNet victim encoder in less than a fifth of the queries required for training.

Proof of Learning (PoL) [22] is a reactive defense that involves the defender claiming ownership of a model by showing incremental updates of the model training. It is a complementary method to dataset inference, which instead identifies a stolen model. PoL could be applied directly to SSL encoders, however, it requires an expensive verification process, where the verifier needs to perform model updates and the prover needs to save intermediate weights of the model, which is more expensive than dataset inference with GMMs. Unfortunately, other current defenses against model stealing for supervised learning are inadequate for defending encoders, and adjusting them to the specificities of encoders is non-trivial [14]. One line of approach is watermarking [9, 14], where the defender embeds a secret trigger into the victim encoder during training to determine ownership at test time. However, watermarking-based defenses have two significant disadvantages. First, researchers have repeatedly shown that adaptive attackers can remove watermarks without severely affecting model performance [7, 21, 40, 44], *e.g.,* through pruning, fine-tuning, rounding or performing backdoor removal [4]. Second, the watermark must be embedded during training; if a model is already trained, or if a watermark defense needs to be updated, the model must be retrained. This is not practical for large encoders.

Another state-of-the-art defense against model stealing in the supervised setting is dataset inference [31] which addresses these disadvantages. However, the adaptation of dataset inference to encoders is difficult, because (1) the algorithm [31] relies on decision boundaries, which do not exist for encoders; and (2) encoders are less prone to overfitting, which provides the signal for dataset inference [14, 31]. Therefore, naive approaches like computing the loss of representations or distances between train and test sets are ineffective [14]. To overcome these issues, we extract more signals from the representations by estimating their densities for train and test sets, as described in Section 3.

When it comes to comparing signals within representations, prior work has considered measuring the similarity between different representations. This has led to the proposal of various similarity metrics including canonical correlation analysis (CCA) [33], centered kernel alignment (CKA) [25], and the orthogonal Procrustes distance [13], which use methods from linear regression, principal component analysis (PCA), and singular value decomposition (SVD). However, these metrics are very general and complex. They have also been shown to disagree in some cases [13]. Since we can only access the final embeddings from encoders, we design metrics more closely related to our setting.

## 3 Defense Method

Dataset inference serves as a defense against model stealing. It enables the model owner or a third-party arbitrator to attribute the ownership of a model in the event that it is stolen. The idea is to take advantage of the effects of knowledge from the victim's training set and to use that as a signature for attributing ownership. Given a well-trained encoder, the effects are small on any single data point; however, when aggregated over many points in the training set, they collectively provide a strong statistical signal for dataset inference. As depicted in Figure 2, for a victim that leverages the private data $D_P$ during training or a stolen copy, we can identify a difference between

distributions of the train data's representations $h_{D_P}$ and the test data's representations $h_{D_N}$ while the distributions of these representations from an independent encoder cannot be distinguished. We use this signal to determine whether a model is a derivative of the victim's training data, *i.e.,* either directly trained on the data or stolen from the victim. To capture the signal, we first partition the victim encoder's training data into two subsets and use one to train Gaussian Mixture Models (GMMs) with the aim of modeling the distribution of representations of data from the encoder's domain. Then, we apply the GMMs to perform dataset inference by measuring the log-likelihood of the encoder's representations of the remaining training data vs some test data (see Figure 1). For the victim encoder or its stolen copies, the log-likelihood of training data representations is significantly higher than the one of test data. We use this to

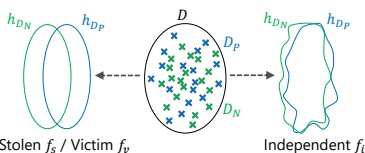

Figure 2: **Dataset Inference - Intuition.** $D_P$ and $D_N$ come from the same distribution. For independent encoders, their representations $h_{D_N}$ and $h_{D_P}$ are i.i.d. while for victim/stolen encoders, $h_{D_N}$ and $h_{D_P}$ induce different distributions.

construct statistical t-tests to determine whether a model is stolen, and present our empirical results in Section 5.

## 3.1 Threat Model

As described in Figure 7, we consider a victim encoder $f_v$ trained on a private training dataset $D_P$. An adversary with black-box access to $f_v$ trains a stolen encoder $f_s$ by querying the victim with data points from a dataset $D_S$ to obtain representations. These representations are then used as part of the training objective for the stolen encoder. During the dataset inference, we assume the presence of a third-party arbitrator, such as law enforcement, with white-box access to the victim's training data, as well as all encoders. Additionally, the arbitrator requires the test dataset $D_N$ from the same distribution as $D_P$ to perform dataset inference. An independent encoder $f_i$ is trained with no access to the victim's private training set $D_P$ and without any queries to the victim's encoder. It is used as a baseline for the ownership resolution in the dataset inference.

## 3.2 Density Estimation of Representations

Representations from encoders contain rich features for given inputs. We analyze the inputs that come from the training and test sets through their representations. For the training inputs, we compute their representations and model their densities [16]. To this end, we leverage GMMs as universal approximators of densities. We give representations a probabilistic interpretation such that they have a smooth enough density which can be approximated by any specific nonzero amount of error using a GMM with enough components. Each component has a separately parameterized mean $\mu$ and covariance $\Sigma$. In some cases, we observe that GMMs can overfit to their training data when no constraints are applied to the covariance matrix, hence we limit the covariance matrix for each component to be diagonal. Moreover, this constraint makes training more computationally efficient since it avoids storing and inverting full high-dimensional covariance matrices.

## 3.3 Data Flow

The full flow of our dataset inference for encoders consists of the following four main steps (which are also visualized in Figure 7 in Appendix):

1. **Victim Training.** The victim's encoder $f_v$ is trained using the whole private training dataset $D_P$.
2. **Encoder Stealing.** To steal the victim encoder $f_v$, an adversary queries $f_v$ with data points from $D_S$ to obtain representations $h_{D_S} \in \mathcal{R}^n : h_{D_S} = f_v(D_S)$. With these representations, the adversary trains the stolen encoder $f_s$ in a contrastive manner.
3. **Training Estimators.** To perform ownership resolution, an arbitrator trains three density estimators $\mathcal{E}_v$, $\mathcal{E}_s$ and $\mathcal{E}_i$ for the victim $f_v$, stolen $f_s$, and independent encoder $f_i$ as follows:
   a) $D_P$ (where $D_P$ is not necessarily the whole private training dataset) is divided into two non-overlapping subsets $D_{P1}$ and $D_{P2}$. While $D_{P2}$ serves as the base for training the density estimators, $D_{P1}$ is used to evaluate density estimates of the private training data vs part of the test data $D_N$.

b) For a given encoder $f \in \{f_v, f_s, f_i\}$, the arbitrator generates representations $h \in \mathcal{R}^n$ : $h_{D_{P2}} = f(D_{P2})$ on dataset $D_{P2}$. Training the density estimators on the respective representations yields the final density estimators $\mathcal{E}_v, \mathcal{E}_s,$ and $\mathcal{E}_i$.

4. **Estimating Densities.** The arbitrator generates representations of $D_{P1}$ and (a subset of) $D_N$ with each encoder $f \in \{f_v, f_s, f_i\}$. Applying the respective density estimator $\mathcal{E} \in \{\mathcal{E}_v, \mathcal{E}_s, \mathcal{E}_i\}$ on the representations yields the log-likelihood of each data point $x$ in the respective dataset: $\forall x \in D : p(x) = \mathcal{E}(f(x))$.

### 3.4 Ownership Resolution

For an encoder $f$, we compute the log-likelihood on $D_{P1}$ and $D_N$ as: $u_P := \frac{1}{|D_{P1}|} \sum_{x \in D_{P1}} \mathcal{E}(f(x))$ and $u_N := \frac{1}{|D_N|} \sum_{x \in D_N} \mathcal{E}(f(x))$. The density estimator $\mathcal{E}$ measures the similarity between the distributions over the victim's representations of the training $D_P$ vs test data $D_N$. The intuition behind the setup is that if an encoder was trained on $D_P$, representations of $D_{P1}$ are much more similar to representations of $D_{P2}$, because the whole dataset $D_P$ was used for training the encoder, however, representations of $D_N$ differ from representation of $D_{P2}$ since $D_N$ was not used to train the encoder. For a victim $f_v$ and a stolen encoder $f_s$, $u_P$ is significantly larger than $u_N$, whereas, for an independent encoder $f_i$, the values do not differ significantly. Finally, we carry out a hypothesis test with the null hypothesis being: $H_0 := u_P \leq u_N$. If the null-hypothesis can be rejected (p-value $< 0.05$), *i.e.,* when the log-likelihood for the training set $D_{P1}$ is higher than that for the test set $D_N$, we can conclude that the tested model was stolen. On the other hand, if the null hypothesis cannot be rejected then the test is inconclusive and we cannot determine if a tested encoder was stolen or not.

## 4 Encoder Similarity Scores

Measuring the quality of stolen encoders allows us to assess attacks and defenses. In standard supervised learning, the quality of a stolen model is evaluated using two main objectives, namely task accuracy, which is the model's performance on the test set, and fidelity, which is the agreement in the predictions for a given task between the stolen and the victim model [19]. One of the approaches to measure the quality of an extracted encoder is to use its outputs to train a downstream task and compute the accuracy of that task or fidelity (with respect to the outputs of the downstream task trained on the victim encoder). However, a single downstream task cannot adequately reflect the degree of similarity between encoders since it reduces their high dimensional embeddings to single label representations, which are confounded by choices of downstream data and training protocol. Instead, we propose two new metrics. Our first metric is an information-theoretic score based on mutual information [10, 27]. Our second metric is a cosine similarity score based on the representations returned by different encoders. These metrics correspond to the *fidelity* metric in supervised learning. The behavior of the two metrics differs in certain cases, for example, when used on obfuscated representations (*e.g.,* with shuffled elements) or with independent models, however, we find that the overall trend is similar. Moreover, the mutual information score is based on an approximation while the cosine similarity score is calculated exactly given representation vectors. Often the effectiveness of defenses may be underestimated against low-quality stolen copies that haven't successfully stolen victim behavior. Our metrics help disentangle such effects and enable faithful evaluation of defenses.

### 4.1 Mutual Information Score

Our first approach to assessing the quality of a stolen encoder uses a score based on mutual information. We sample $N$ data points from the victim's private training dataset $D_P$ and pass them through the encoders $f_v, f_s,$ and $f_i$ to generate the respective representations. Per standard practice, we recenter and normalize the representations [13]. We denote the entropy by $H$ and compute it according to Algorithm 2 which takes $f_v$, $D_P$, and $N$ as input. For the joint entropy $H(f_v, f_s|D)$, we generate representations from the two encoders (in this case victim $f_v$ and stolen $f_s$) and concatenate them, which increases the dimensionality of the final representation to $2d$, while other steps remain unchanged. A detailed algorithm for computing the joint entropy can be found in the Appendix as Algorithm 3. We compute an **approximate score** that is based on the definition of mutual information $I(f_v, f_s|D_P)$ between the victim encoder $f_v$ and the stolen copy $f_s$ as well as the analogous mutual information $I(f_v, f_i|D_P)$ between the victim encoder $f_v$ and the independently trained encoder $f_i$.

We rely on approximations since we measure mutual information using finite data. Yet, in practice, such approximations have proven useful [32]. We define our mutual information score as follows:

$$I(f_v, f_s|D_P) = H(f_v|D_P) + H(f_s|D_P) - H(f_v, f_s|D_P). \tag{1}$$

A higher value of the mutual information $I(f_v, f_s|D_P)$ indicates a higher information leakage incurred by the stolen encoder. Expectedly, mutual information is higher between the victim and the stolen encoder than between the victim and independent encoders $I(f_v, f_s|D_P) >> I(f_v, f_i|D_P)$. We can normalize mutual information into a score (between 0 and 1) by setting the lower bound as the mutual information between the victim $f_v$ and a randomly initialized model $f_r$: $I_{min} = I(f_v, f_r|D_P)$ and the upper bound as the mutual information between the victim and itself: $I_{max} = I(f_v, f_v|D_P)$ For the current mutual information score $I_c$, the normalized score is defined as $S := \frac{I_c - I_{min}}{I_{max} - I_{min}}$.

## 4.2 Cosine Similarity Score

The second score we use to assess the quality of a stolen encoder is based on the cosine similarity between its representations and the victim's representations. More specifically, we first compute representations for the two encoders on a set of $N$ randomly selected data points from the dataset $D_P$. Again as per standard practice [13], we recenter and normalize these representations. For each of the $N$ inputs, we then compute the cosine similarity between the corresponding representations from both encoders where the cosine similarity $\text{sim}(a, b) = \frac{a^T b}{||a||_2 ||b||_2}$ for representation vectors $a$ and $b$.

We show (in Section 4.3) that the loss functions, which are used for stealing encoders, directly maximize the cosine similarity between representations from victim and stolen encoders. We thus propose to use the cosine similarity score $C$ as a metric, which we define as: $C = |\text{sim}(a, b)|$ (2). The score yields values in the range $[0, 1]$, with a higher score indicating closer representations. To calculate a per-encoder cosine similarity score, we average the cosine similarity scores over all inputs. We find that the cosine similarity score is well-calibrated across encoders. Namely, an independent encoder, expected to have representations unrelated to the victim encoder, has an average cosine similarity concentrated around 0 [9], while a stolen encoder exhibits significantly higher scores. The cosine similarity score is also easy to compute since it only requires the corresponding representations of the two models and their dot product.

## 4.3 Analysis

There are various ways in which an attacker may steal an encoder. To simplify our analysis of the cosine similarity score, we consider the two best-performing loss functions used for stealing [14]: the first where the attacker minimizes the non-contrastive MSE (Mean Squared Error) loss between its representations and the victim encoder's representations to train the stolen encoder, and the second where the attacker uses a contrastive loss function, such as the InfoNCE loss [43] which is used in SimCLR [5].

**Stealing with MSE loss.** In the case where the MSE loss is used, let $x_i$ be a query made by an attacker and let $f_v(x_i) = h_{v_i}, f_s(x_i) = h_{s_i} \in \mathcal{R}^n$ be the corresponding representations of the victim and stolen encoders, respectively. The MSE loss between these two representations is $\frac{1}{n}\sum_{j=1}^{n}(h_{v_{ij}} - h_{s_{ij}})^2 = \frac{1}{n}||h_{v_i} - h_{s_i}||_2^2$. It follows directly that minimizing the MSE loss also minimizes the $\ell_2$ distance between representations and equivalently maximizes the cosine similarity between representations: **Theorem 1** $||a - b||_2 = \sqrt{2(1 - \text{sim}(a, b))}, ||a||_2 = ||b||_2 = 1$ (see G.1).

**Stealing with a contrastive loss.** When an attacker uses a contrastive loss function for stealing, minimizing the loss corresponds to maximizing the sum of the cosine similarities between positive pairs, *i.e.*, $\sum_{c=1}^{m}(\text{sim}(h_{s_c}, h_{v_c})/\tau)$. The InfoNCE loss, or contrastive losses in general, also increase the mutual information score [3, 43]. We therefore expect that stolen encoders will have larger similarity scores w.r.t. the victim encoder than independent encoders. We refer the reader to Appendix G.1 for a more detailed discussion of the loss functions and their relationship with the similarity scores.

# 5 Empirical Evaluation

We evaluate our defense against encoder extraction attacks using five different vision datasets (CIFAR10, CIFAR100 [28], SVHN [34], STL10 [8], and ImageNet [11]). Table 1 shows that our dataset inference method is able to differentiate between the stolen copies of the victim encoder and independently trained encoders by using the victim's private training data as the signature. We also show that our defense works in the scenario where the adversary modifies the representations to render them inconspicuous, *e.g.,* by shuffling the order of elements in the representation vectors. To assess the quality of the stolen encoders and the performance of our defense, we measure the mutual information and cosine similarity scores between encoders and present our results in Tables 2 and 3.

## 5.1 Training Victim, Stolen and Independent Encoders

**Victim.** We use victim encoders trained on the ImageNet, CIFAR10, and SVHN datatsets. For the ImageNet victim encoder , we use a model released by the authors of SimSiam [6]. To train CIFAR10 and SVHN victim encoders, we use an open-source PyTorch implementation of SimCLR [3]. For SVHN, we merge the original training and test splits, and use the randomly-selected 80% as the training set and the rest 20% as the test set. This is necessary because the original training and test splits for SVHN are not i.i.d [36], which violates the assumption for dataset inference (see Section B). The ImageNet victim has an output representation dimension of 2048, while the CIFAR10 and SVHN victim encoders have 512-dimensional representations.

**Stolen.** When stealing from the victim encoders, we evaluate different numbers of queries from various datasets, including CIFAR10, SVHN, ImageNet, and STL10. Stolen encoders are trained in a similar contrastive way as the victim and use the InfoNCE loss, where the positive pairs consist of representations from the victim and stolen encoder for a given input. Algorithm 1 summarizes the stealing approach used by an adversary.

**Independent.** For each victim encoder, we train independent encoders using datasets different from the victim's private training dataset $D_P$. The encoders are trained with the SimCLR approach, similar to the way the victim encoders were trained. In the case where the dataset used to train the independent model had different image dimensions from the victim's training dataset, the dataset was resized to be of the same size.

More details on the training and stealing of encoders can be found in Section D.3 of the Appendix.

## 5.2 Dataset Inference on Encoders

**Setup.** We train GMMs with 10 components for SVHN and CIFAR10, and 50 components for ImageNet. In general, we observe that the larger number of components for GMMs, the better the defense is. For ImageNet, we restrict the covariance matrix to be diagonal for efficiency. For CIFAR10 and SVHN, we use the full covariance matrix. For SVHN and CIFAR10, we use 50% of the training set to train GMMs, and the remaining for evaluation. For ImageNet, we use $100K$ images from the training set to train GMMs, and another $100K$ of the training set as an evaluation set. We normalize representations by $l_2$ norm for training GMM. For ImageNet, we also standardize representations (subtract mean and divide by standard deviation) before normalization. We do not use augmentations in dataset inference. For each setting, the hyperparameters are tuned on the victim model and a randomly-initialized model.

**Evaluation of our Defense.** The empirical results in Table 1 demonstrate that we are able to differentiate between stolen and independent encoders from the difference in log-likelihoods. We observe that the stolen encoders have significantly larger $\Delta\mu$ than the independent encoders. The p-values further show that for stolen encoders the null hypothesis is rejected while for independent encoders, the test is inconclusive. Similar to dataset inference for supervised learning [31], the victim model typically has the largest $\Delta\mu$ and the smallest p-values. We also observe that our method is better at detecting encoders that are stolen using queries from the victim's training set.

**Number of Stolen Queries.** Table 3 shows that as the attacker steals with more queries, the p-value from our defense becomes lower. This is consistent with the finding in [31] that dataset inference works better with stronger stolen encoders. We also find that our defense is able to detect stolen

---

[3]https://github.com/kuangliu/pytorch-cifar

Table 1: **Dataset inference via density estimation of representations.** We detect if a given encoder was stolen. $f_v$ denotes the victim encoder trained on data $D$, $f_s$ is the stolen encoder extracted using queries from a given stealing dataset $D$, and $f_i$ is an independent encoder trained on data $D$ (different than the victim's private training data). Each value is an average of 3 trials. $\Delta\mu$ is the effect size from the statistical t-test. Obfuscations: the representation can be modified by an attacker in the following ways: (1) *Shuffle* the elements in the representation vectors, (2) *Pad* with zeros or add zeros at random positions, and (3) apply a linear *Transform*. The first row below denotes the victim's private data $D_P$.

| Victim's private data: | | | CIFAR10 | | | SVHN | | | ImageNet | |
|---|---|---|---|---|---|---|---|---|---|---|
| Encoder | Obfuscate | $D$ | p-value | $\Delta\mu$ | $D$ | p-value | $\Delta\mu$ | $D$ | p-value | $\Delta\mu$ |
| $f_v$ | N/A | CIFAR10 | 5.61e-82 | 18.92 | SVHN | 2.75e-125 | 23.88 | ImageNet | 6.23e-14 | 7.09 |
| $f_s$ | N/A | SVHN | 3.97e-2 | 3.04 | SVHN | 6.35e-41 | 13.36 | SVHN | 3.33e-4 | 4.04 |
| | | CIFAR10 | 8.73e-7 | 5.09 | CIFAR10 | 2.38e-4 | 4.61 | CIFAR10 | 1.47e-4 | 6.21 |
| | | STL10 | 1.04e-2 | 3.42 | STL10 | 1.23e-5 | 5.22 | STL10 | 1.09e-4 | 5.87 |
| | | ImageNet | 6.34e-3 | 3.47 | ImageNet | 9.81e-3 | 3.74 | ImageNet | 3.14e-5 | 7.32 |
| $f_s$ | Shuffle | CIFAR10 | 1.72e-6 | 4.98 | CIFAR10 | 7.32e-4 | 4.77 | CIFAR10 | 6.72e-4 | 5.21 |
| | Pad | CIFAR10 | 3.44e-6 | 4.84 | CIFAR10 | 2.51e-3 | 3.08 | CIFAR10 | 2.31e-3 | 4.23 |
| | Transform | CIFAR10 | 6.81e-7 | 5.11 | CIFAR10 | 6.45e-3 | 3.32 | CIFAR10 | 8.45e-3 | 3.98 |
| $f_i$ | N/A | CIFAR100 | 3.67e-1 | -0.37 | CIFAR100 | 6.21e-1 | 0.52 | CIFAR100 | 7.53e-2 | 1.63 |
| | | SVHN | 2.96e-1 | 0.98 | CIFAR10 | 4.82e-1 | 0.56 | SVHN | 5.42e-1 | 0.69 |

encoders even if the attacker only steals from a small number of queries. For example, in Table 3, we are able to claim ownership when only 50K - 100k queries are used for stealing ImageNet victims (around 4% of its training set).

**Robustness of Dataset Inference to Obfuscations.** The attacker can obfuscate the stolen encoder representations by, for instance, applying shuffling (changing the order of elements), padding (adding zeros), or linear transformations (*e.g.,* scaling or adding a constant). These obfuscations have little impact on the downstream performance [17] but may pose challenges to the defenses of the victim. The results in Table 1 show that the p-values for the stolen encoders after attackers' obfuscations remain low, which implies that our method is robust to these types of obfuscations.

### 5.3 Measuring Quality of Stolen Encoders

**Setup.** To measure the quality of stolen encoders, we select a random subset of $N = 20K$ unaugmented images from the private training dataset $D_P$ and compute their representations from stolen and victim encoders. We then centralize (subtract the mean for each dimension) and normalize the representations (divide by the $\ell_2$ norm). For the mutual information score, we first estimate the entropies $H(f_v), H(f_s), H(f_v, f_s)$, which are then added and normalized as in Section 4.1. The score is capped to be in the range $[0, 1]$. To compute the cosine similarity score, we find the absolute value of the dot product of corresponding representations for the two encoders (Equation 2). These dot products are then averaged over all representations.

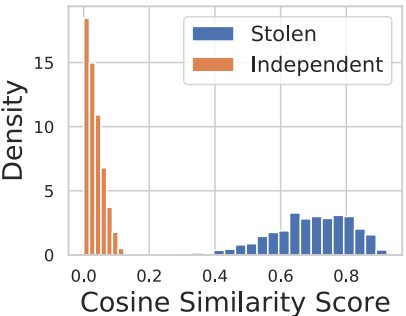

Figure 3: Distribution of **cosine similarity scores**.

**Evaluation of Metrics.** To evaluate the mutual information and the cosine similarity scores, we conduct two sets of experiments to verify if: (1) the scores are higher for stolen than independent encoders, and (2) the scores increase as more queries are used to steal encoders, which suggests a higher quality of the stolen copies [14]. In Table 2, we observe that both our scores assign higher values to the stolen encoders than the independent encoders. Table 3 shows that our mutual information and cosine similarity scores generally increase while the p-values from our dataset inference decrease with respect to the number of queries used to steal an encoder. This implies that the performance of our defense is consistent with the similarity metrics and becomes more effective as the quality of the stolen encoder improves. We also plot a histogram of the cosine similarity scores for the stolen and independent encoders in Figure 3 for an SVHN victim encoder, a stolen encoder from it (using CIFAR10 training data for queries), and an independent encoder (trained on CIFAR100). There is a pronounced difference between the two distributions with the

Table 2: **Encoder similarity scores.** We compare encoders via the encoder quality metrics using the same setting as in Table 1. We compute the score $S(\cdot, f_v)$ based on the mutual information between a given encoder (in a row) and the victim encoder $f_v$. Analogously, we compute the cosine similarity score $C(\cdot, f_v)$.

| Victim's private data: | | | CIFAR10 | | | SVHN | | | ImageNet | |
|---|---|---|---|---|---|---|---|---|---|---|
| Encoder | Obfuscate | $D$ | $S(\cdot, f_v)$ | $C(\cdot, f_v)$ | $D$ | $S(\cdot, f_v)$ | $C(\cdot, f_v)$ | $D$ | $S(\cdot, f_v)$ | $C(\cdot, f_v)$ |
| $f_v$ | N/A | CIFAR10 | 1.0 | 1.0 | SVHN | 1.0 | 1.0 | ImageNet | 1.0 | 1.0 |
| | N/A | SVHN | 0.73 | 0.504 | SVHN | 0.96 | 0.91 | SVHN | 0.86 | 0.39 |
| $f_s$ | N/A | CIFAR10 | 0.84 | 0.95 | CIFAR10 | 0.94 | 0.69 | CIFAR10 | 0.88 | 0.43 |
| | N/A | STL10 | 0.89 | 0.92 | STL10 | 0.95 | 0.89 | ImageNet | 0.96 | 0.78 |
| | *Shuffle* | SVHN | 0.74 | 0.002 | CIFAR10 | 0.94 | 0.003 | SVHN | 0.86 | 0.005 |
| $f_s$ | *Pad* | SVHN | 0.74 | 0.007 | CIFAR10 | 0.93 | 0.013 | SVHN | 0.85 | 0.003 |
| | *Transform* | SVHN | 0.75 | 0.504 | CIFAR10 | 0.93 | 0.69 | SVHN | 0.86 | 0.39 |
| $f_i$ | N/A | CIFAR100 | 0.63 | 0.0007 | CIFAR100 | 0.90 | 0.007 | CIFAR100 | 0.81 | 0.0018 |
| | N/A | SVHN | 0.12 | 0.0001 | CIFAR10 | 0.90 | 0.009 | SVHN | 0.75 | 0.002 |

Table 3: **Encoder similarity scores and p-values from dataset inference vs the number of queries.** The quality of the stolen encoders increases with more stealing queries, which is reflected by the rise in the mutual information and cosine similarity scores as well as the better performance of our defense as indicated by the decreasing p-values. $D_P$ is the private dataset used to train the victim and $D_S$ is the dataset used for stealing.

| $D_P$ | $D_S$ | Score | Number of Queries | | | | | | | | |
|---|---|---|---|---|---|---|---|---|---|---|---|
| | | | 5K | 10K | 20K | 30K | 40K | 50K | 100K | 200K | 250K |
| | | $S(\cdot, f_v)$ | 0.62 | 0.79 | 0.79 | 0.81 | 0.82 | 0.84 | 0.85 | 0.85 | 0.86 |
| ImageNet | SVHN | $C(\cdot, f_v)$ | 0.25 | 0.32 | 0.33 | 0.36 | 0.35 | 0.38 | 0.38 | 0.40 | 0.39 |
| | | p-values | 1.23e-1 | 7.91e-2 | 6.53e-2 | 8.98e-2 | 4.52e-2 | 1.10e-2 | 2.11e-3 | 1.11e-3 | 3.33e-4 |
| | | | 500 | 5K | 7K | 8K | 9K | 10K | 30K | 40K | 50K |
| | | $S(\cdot, f_v)$ | 0.55 | 0.60 | 0.62 | 0.75 | 0.58 | 0.64 | 0.87 | 0.82 | 0.88 |
| ImageNet | CIFAR10 | $C(\cdot, f_v)$ | 0.21 | 0.28 | 0.31 | 0.29 | 0.36 | 0.32 | 0.40 | 0.41 | 0.43 |
| | | p-values | 8.88e-2 | 7.12e-2 | 8.23e-1 | 4.14e-1 | 3.41e-3 | 8.51e-21 | 9.23e-2 | 7.32e-2 | 1.47e-4 |
| | | | 500 | 5K | 7K | 8K | 9K | 10K | 30K | 50K | 100K |
| | | $S(\cdot, f_v)$ | 0.76 | 0.75 | 0.72 | 0.81 | 0.84 | 0.81 | 0.89 | 0.88 | 0.92 |
| ImageNet | STL10 | $C(\cdot, f_v)$ | 0.28 | 0.29 | 0.36 | 0.38 | 0.37 | 0.43 | 0.44 | 0.52 | 0.58 |
| | | p-values | 9.63e-1 | 8.21e-1 | 7.32e-1 | 5.44e-1 | 1.21e-1 | 5.98e-2 | 8.11e-2 | 6.28e-2 | 1.09e-4 |
| | | | 5K | 10K | 20K | 30K | 40K | 50K | 100K | 200K | 250K |
| | | $S(\cdot, f_v)$ | 0.61 | 0.75 | 0.73 | 0.76 | 0.81 | 0.91 | 0.90 | 0.95 | 0.96 |
| ImageNet | ImageNet | $C(\cdot, f_v)$ | 0.29 | 0.48 | 0.49 | 0.51 | 0.46 | 0.38 | 0.52 | 0.76 | 0.78 |
| | | p-values | 9.88e-1 | 3.21e-1 | 5.32e-1 | 1.08e-1 | 3.61e-3 | 3.97e-4 | 5.34e-4 | 8.72e-4 | 3.14e-5 |

cosine similarity scores for the independent encoder being close to $0$ and the scores for the stolen encoder being much higher than $0$.

**Robustness of Metrics to Obfuscations.** We also consider the effect of obfuscations on these metrics. Without any obfuscation of the representations from stolen encoders, the cosine similarity score shows a clearer distinction between stolen and independent encoders than the mutual information score: in Table 2, the cosine similarity scores for all independent encoders are close to zero, but the mutual information scores can be quite high (such as $0.9$ for the independent encoders of SVHN, which is likely because of the mutual information score being based on an approximation). However, the mutual information score is robust to the obfuscations of the attackers while cosine similarity is not: in Table 2, the cosine similarity score for the stolen encoders after shuffling and padding drops close to zero. Mutual information, as a more general metric based on the information measurement instead of the brittle structure of the representation vectors, performs better and is oblivious to the obfuscations that attackers might introduce.

## 5.4 Limitations

If the t-test run as part of dataset inference is inconclusive for an extracted encoder, we cannot state whether the encoder was stolen. Similarly, for an independent encoder, there is the possibility of it being incorrectly classified as stolen. Previous work [26, 38] has shown that self-supervised encoders trained using heavy augmentations and contrastive learning generalize better than their supervised counterparts, which makes it harder for the dataset inference to differentiate between train and test representations in SSL than in the SL setting [14]. The loss values of projected individual

representations are insufficient for dataset inference [14]. We build on top of this observation to enable dataset inference for encoders and use GMMs to distinguish between train and test representations.

## 6  Conclusions

New public APIs expose self-supervised encoder models which return high-dimensional embeddings for provided inputs. Adversaries can use these embeddings to steal the encoders. We present a novel method based on dataset inference for defending against such stealing attacks along with metrics to assess the quality of the stolen encoders and to quantify the effectiveness of our defense. We observe that knowledge contained in the private training set is transferred from the victim encoder to its stolen copy. Thus, the private data acts as a signature of the victim encoder. By leveraging density estimation on the respective encoders' representations, we obtain a signal allowing us to differentiate between the encoder's training and test data. This difference is detectable in both the victim encoder and its stolen copy but not in independent encoders which are legitimately trained on different data than the victim's private training data. Thus, we are able to flag the stolen copy of the victim encoder while not accusing creators of legitimately trained encoders of theft. We show the high effectiveness of our defense on vision encoders. Future work may explore additional applications of our proposed defense and metrics beyond model stealing and ownership verification, as well as their use in other domains such as natural language processing (NLP). In particular, our method may help enforce the ethical usage of sensitive online data, such as images on social media, in accordance with privacy regulations by auditing if a given provider's encoder contains knowledge of these sensitive data.

## Acknowledgments

We would like to acknowledge our sponsors, who support our research with financial and in-kind contributions: CIFAR through the Canada CIFAR AI Chair program, DARPA through the GARD program, Intel, Meta, NFRF through an Exploration grant, and NSERC through the Discovery Grant and COHESA Strategic Alliance. Resources used in preparing this research were provided, in part, by the Province of Ontario, the Government of Canada through CIFAR, and companies sponsoring the Vector Institute. We would like to thank members of the CleverHans Lab for their feedback.

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
