# OpenReview forum: "Dataset Inference for Self-Supervised Models"
_NeurIPS.cc/2022/Conference — NeurIPS 2022 Accept_

### Official Review · Reviewer_Ay7D · 2022-07-06

**Rating:** 7
**Confidence:** 4
**Soundness:** 4 excellent
**Presentation:** 4 excellent
**Contribution:** 3 good

**Summary:**

The paper proposes a new defense against model stealing of self-supervised models, using the recently proposed dataset inference framework. Similar to original dataset inference for supervised models, the proposed defense detects stealing based on a hypothesis that stolen models exhibit different behavior on private and test data. In this case, this is quantified by using auxilliary density estimation models and observing the log likelihood of data from the two sources. Further, the paper proposes two metrics to measure the quality of stealing: mutual information score and cosine similarity score.

**Questions:**

- The motivation for the additional contribution (two scores) is somewhat unclear. Can the authors make a stronger point for why linear evaluation on several tasks (as in Table 3 in [14]) is insufficient/unnecessary; what is the reason for not including such evaluation alongside proposed metrics? In a similar vein, both scores (MI and cosine) to me seem analogous to fidelity, and not accuracy as claimed in the paper. Can the authors elaborate on this claim further? Making this part more cohesive with the rest of the paper would make the contribution stronger.
- Some results seem to be missing in Table 1, e.g. ImageNet as $D_S$ for first two choices of $D_P$. While the conclusions certainly hold in almost all cases, the test does not pass for CIFAR10 as $D_P$ and SVHN as $D_S$, making it more necessary to provide a complete and convincing evaluation.
- Further, 4.3 claims that the paper focuses on both MSE and InfoNCE stealing losses, but 5.1 seems to be imply only InfoNCE is used in the evaluation. Can the authors comment on this mismatch? The evaluation would be stronger if MSE and SoftNN were added.
- (Nit) It would be interesting to see if different density estimation methods would be able to further boost the performance, did the authors experiment with this?

**Limitations:**

I find the discussion of societal impact sufficient. Regarding limitations, it would be good to more explicitly acknowledge the strength of results compared to the supervised case and reflect on statements given in [14]. More importantly, to avoid the cat-and-mouse nature of attack/defense work in the field (similar to adversarial examples literature until recently), I find it necessary to consider the prospect of adaptive adversaries, i.e. the robustness of the defense against attackers aware of the proposed defense. I acknowledge that designing an adaptive attacker is often a highly non-trivial contribution in itself; a short explicit discussion would still benefit the paper.

**Strengths And Weaknesses:**

The paper is a solid contribution to an active field and is a good follow-up to [14] which posit that dataset inference is significantly harder in the case of SSL. While the results still are less convincing than in the supervised case, the work clearly demonstrates that successfully adapting dataset inference to SSL is possible if more tailored approaches are used. The paper is very well written and structured.

I believe the work has no fundamental weaknesses. However, I have several relatively minor concerns (raised below in Questions), which I hope the authors can sufficiently address in their response; I am willing to reassess the work after the discussion.

---

> ### Author Response · Authors · 2022-08-02
> **Limitations and an adaptive adversary**
>
> >**Regarding limitations, it would be good to more explicitly acknowledge the strength of results compared to the supervised case and reflect on statements given in [14].**
>
> Indeed, the results from the Supervised Learning (SL) show that the detection of the stolen model is possible with fewer queries than in the SSL setting and the significance level in SL is set to 1% while it is 5% for SSL. Previous work [1, 2] have shown that self-supervised encoders trained using heavy augmentations and contrastive learning generalize better than their supervised counterparts, which makes it harder for the dataset inference to differentiate between train and test representations in SSL than in SL setting.
>
> The dataset inference is harder to apply in the SSL setting than in the supervised setting [14]. The preliminary work [14] assumed access to the projection head, which usually is not accessible for public APIs that return only representations instead of their projections. The method based on the loss values from [14] considers the data points separately. We directly use representations instead of their projections and extend the method to model the distributions of train vs test representations holistically. We found that using GMMs allowed us to differentiate between the representations from train vs test sets, which was not possible by utilizing the loss values of projected representations.  We added Section 5.4 on Limitations in the main paper.
>
> *References:*
>
> [1] [Concept Generalization in Visual Representation Learning](https://openaccess.thecvf.com/content/ICCV2021/html/Sariyildiz_Concept_Generalization_in_Visual_Representation_Learning_ICCV_2021_paper.html). Mert Bulent Sariyildiz, Yannis Kalantidis, Diane Larlus, Karteek Alahari. ICCV 2021.
>
> [2] [Contrasting Contrastive Self-Supervised Representation Learning Pipelines.](https://openaccess.thecvf.com/content/ICCV2021/html/Kotar_Contrasting_Contrastive_Self-Supervised_Representation_Learning_Pipelines_ICCV_2021_paper.html) Klemen Kotar, Gabriel Ilharco, Ludwig Schmidt, Kiana Ehsani, Roozbeh Mottaghi. ICCV 2021.
>
> [14] [On the difficulty of defending self-supervised learning against model extraction.](https://proceedings.mlr.press/v162/dziedzic22a.html) Adam Dziedzic, Nikita Dhawan, Muhammad Ahmad Kaleem, Jonas Guan, Nicolas Papernot. ICML 2022.
>
> >**I find it necessary to consider the prospect of adaptive adversaries.**
>
> We introduce the possible adaptive attacks against our method as obfuscations (shuffle, pad, transform). We show that our method can still help detect model stealing even when the adversary obfuscates their representations. One other obfuscation method could pad the representations with noise values instead of zeros. Then, we should expect that the more noise (e.g., higher standard deviation for the Gaussian noise) we add, the worse the performance of our defense. However, this would also lower the downstream accuracy of the stolen model, which gives the trade-off between the strength of the adaptive attack and the utility of the stolen encoder. Additionally, the shuffle and pad obfuscations can hinder the performance of the cosine similarity score in assessing the quality of stolen encoders but the mutual information score is a more robust metric in these cases.
>
> We ran additional experiments for random obfuscation. We add Gaussian noise $N(0, 0.1)$ to the random dimensions of representations. Below, we present the mutual information scores (denoted as S) and p-values for the CIFAR10 victim encoder stolen using the STL10 dataset. In the first (header) row *Corrupt %* represents the percent of representation dimensions being obfuscated. 0 means no obfuscation, and 100 means all dimensions are corrupted.
>
> | Corrupt % | 0       | 20     | 40     | 60     | 80     | 100       |
> |-----------|---------|---------|---------|---------|---------|---------|
> | S         | 0.89    | 0.81    | 0.77    | 0.76    | 0.71    | 0.69    |
> | p-value   | 1.04e-2 | 3.76e-2 | 4.91e-2 | 8.00e-2 | 1.03e-1 | 4.73e-1 |

---

> ### Author Response · Authors · 2022-08-02
> **Loss functions for stealing and different density estimation methods**
>
> >**Further, 4.3 claims that the paper focuses on both MSE and InfoNCE stealing losses, but 5.1 seems to be imply only InfoNCE is used in the evaluation. Can the authors comment on this mismatch? The evaluation would be stronger if MSE and SoftNN were added.**
>
> We focused on the InfoNCE loss for stealing since, in most cases, it gave the best performance of the stolen encoders. For completeness, we report the results of our ownership resolution for different loss functions used for stealing (as in Table 2 in [14]). We steal the CIFAR10 encoder using 9000 queries from the CIFAR10 test data with the MSE, InfoNCE and SoftNN loss functions. As the table below shows, we are able to achieve low p-values for all of the loss functions.
>
> | Loss | Accuracy (%) on CIFAR10 | p-value | $\Delta \mu$ (effect size) |
> |---------|---------------|--------------------|--------------|
> | *Victim* |  86.9   | 1.37e-16    | 9.34     |
> | MSE | 79.8  |  7.82e-11      | 8.39    |
> | InfoNCE |  77.6  | 3.56e-6  | 4.64 |
> | SoftNN | 77.8  |      5.64e-4  | 3.98    |
>
> >**(Nit) It would be interesting to see if different density estimation methods would be able to further boost the performance, did the authors experiment with this?**
>
> We chose GMMs because other density estimation models such as normalizing flows are overparameterized and would require more training data. In essence, we do not need to model the density of the representations very accurately: our only requirement is for the method to distinguish between the densities of the representations.

---

> ### Author Response · Authors · 2022-08-02
> **Fidelity scores and results in Table 1**
>
> >**In a similar vein, both scores (MI and cosine) to me seem analogous to fidelity, and not accuracy as claimed in the paper. Can the authors elaborate on this claim further? Making this part more cohesive with the rest of the paper would make the contribution stronger.**
>
> We agree that mutual information and cosine similarity scores are more similar to the fidelity score from supervised learning instead of accuracy. We indicated this in the abstract: *“As part of our evaluation, we also propose to measure the fidelity of stolen encoders without involving downstream tasks and to quantify the effectiveness of the theft detection; instead, we leverage mutual information and distance measurements.”* We corrected the statement at the beginning of Section 4 in the revised version of the paper.
>
> >**Some results seem to be missing in Table 1, e.g. ImageNet as D_S for first two choices of D_P. While the conclusions certainly hold in almost all cases, the test does not pass for CIFAR10 as D_P and SVHN as D_S, making it more necessary to provide a complete and convincing evaluation.**
>
> We ran additional experiments to steal from SVHN and CIFAR10 victim encoders using ImageNet queries. The results for running dataset inference on these stolen models are as follows:
> | Victim’s Dataset | Stealing Dataset | p-value | $\Delta \mu$ (effect size) |
> |---------|---------------|--------------------|--------------|
> | CIFAR10    |  ImageNet      |  6.34e-3          |3.47 |
> | SVHN      | ImageNet          | 5.32e-8    | 5.52    |
>
> We updated Table 1 to include the required results (e.g., ImageNet as D_S for the first two choices of D_P).  We also improved the GMM model (increased the number of components from 10 to 20) and obtained higher confidence (p-value below 0.05) that the encoder was indeed stolen for the pointed out case with CIFAR10 as $D_P$ and SVHN as $D_S$ (the victim model trained on CIFAR10 and stealing with the SVHN dataset).
>
> |        | Dataset  | Obfuscation | p-value  | $\Delta \mu$ (effect size) |
> |--------|----------|-------------|----------|----------|
> | Victim | CIFAR10  |             | 4.52e-17 | 10.73    |
> | Stolen | SVHN     |             | 3.97e-2  | 3.04     |
> | Stolen | CIFAR10  |             | 8.73e-7  | 5.09     |
> | Stolen | STL10    |             | 1.04e-2  | 3.42     |
> | Stolen | CIFAR10  | shuffle     | 1.72e-6  | 4.98     |
> | Stolen | CIFAR10  | pad         | 3.34e-6  | 4.84     |
> | Stolen | CIFAR10  | transform   | 6.81e-7  | 5.11     |
> | Independent | CIFAR100 |             | 3.67e-1  | -0.37    |
> | Independent | SVHN     |             | 2.96e-1  | 0.98     |

---

> ### Author Response · Authors · 2022-08-02
> **Additional scores**
>
> We thank the reviewer for the valuable, detailed, and positive feedback. Our responses are in-line below:
>
> >**The motivation for the additional contribution (two scores) is somewhat unclear. Can the authors make a stronger point for why linear evaluation on several tasks (as in Table 3 in [14]) is insufficient/unnecessary; what is the reason for not including such evaluation alongside proposed metrics?**
>
> Our motivation is to improve the assessment of the quality of stolen encoders by providing complementary evaluation metrics, in addition to downstream accuracy. Relying on downstream accuracy is often insufficient in our setting, because an independently trained encoder can have high downstream accuracy without being stolen from the victim encoder. An example of this phenomenon is shown in the results below (from Table 9): independently trained encoders have high downstream accuracy, but low cosine similarity.
>
> |   Number of Queries    |   Dataset for Stealing |Downstream  CIFAR10   | Downstream STL10 | Downstream SVHN | Cosine Similarity |
> |-------------|----------------|----------------|------|------|-----------------|
> |500          | CIFAR10        |24.5            |23.3  |19.3  |0.24             |
> |5K           | CIFAR10        |53.3            |45.2  |58.1  |0.4              |
> |7K           | CIFAR10        |57.9            |50.9  |69.2  |0.46             |
> |8K           | CIFAR10        |59.6            |52    |72.6  |0.47             |
> |9K           | CIFAR10        |59.9            |50.8  |71.5  |0.49             |
> |10K          | CIFAR10        |59.1            |51.3  |72.3  |0.52             |
> |20K          | CIFAR10        |60.6            |51.5  |73.4  |0.58             |
> |30K          | CIFAR10        |60.7            |52.1  |74.1  |0.63             |
> |50K          | CIFAR10        |59.8            |51.6  |75.1  |0.69             |
> |50K          | STL10          |60.6            |51.5  |76.8  |0.89             |
> |50K          | SVHN           |55.6            |48.7  |82.2  |0.91             |
> |*Independent*| CIFAR10        |87.4            |73.4  |49.5  |0.009            |
> |*Independent*| CIFAR100       |73.8            |61.7  |67.7  |0.007            |

---

> > ### Author Response · Authors · 2022-08-02
> > **New results from Table 10 for the comparison between the accuracy on downstream tasks vs new metrics**
> >
> > Next, we also present our new results from Table 10. For example, we observe that when stealing with SVHN queries from the CIFAR10 victim encoder, we cannot identify the stolen encoder as extracted from the victim based on the victim's performance on downstream tasks. On the other hand, our metrics show evidence of stealing after around 30K queries. Additionally, our metrics correctly show a negligible similarity between the victim and independent encoders. We also note that the cosine similarity score has an almost linear dependence on the number of queries that the adversary used to extract the victim encoder, which is helpful to adequately measure the quality of this stolen encoder.
> >
> > | \# of Queries  | Dataset  | CIFAR10 | STL10 | SVHN | $S(\cdot,f_v)$ | $C(\cdot,f_v)$ | p-value  |
> > |----------------|----------|---------|-------|------|----------------|----------------|----------|
> > | Victim Encoder | N/A      | 87.4    | 73.4  | 49.5 | 1              | 1              | 1.37e-16 |
> > | 500            | SVHN     | 22.3    | 20.8  | 20.1 | 0.00           | 0.07           | 5.69e-1  |
> > | 1K             | SVHN     | 26.5    | 23.4  | 29.9 | 0.06           | 0.11           | 9.02e-1  |
> > | 2K             | SVHN     | 40.0    | 35.2  | 46.7 | 0.12           | 0.13           | 7.83e-1  |
> > | 3K             | SVHN     | 44.4    | 39.8  | 56.3 | 0.18           | 0.21           | 8.04e-1  |
> > | 4K             | SVHN     | 47.4    | 42.5  | 60.8 | 0.22           | 0.19           | 6.30e-1  |
> > | 5K             | SVHN     | 49.5    | 43.8  | 60.9 | 0.29           | 0.28           | 4.28e-1  |
> > | 10K            | SVHN     | 55.6    | 46.6  | 58.9 | 0.38           | 0.36           | 6.42e-1  |
> > | 20K            | SVHN     | 57.7    | 48.7  | 60.4 | 0.43           | 0.39           | 9.81e-2  |
> > | 30K            | SVHN     | 58.0    | 49.2  | 57.0 | 0.57           | 0.45           | 6.73e-2  |
> > | 40K            | SVHN     | 61.4    | 51.1  | 55.0 | 0.69           | 0.49           | 4.83e-2  |
> > | 50K            | SVHN     | 61.2    | 51.7  | 54.8 | 0.76           | 0.50           | 3.97e-2  |
> > | 50K            | CIFAR10  | 84.8    | 70.7  | 52.4 | 0.84           | 0.95           | 8.73e-7  |
> > | 50K            | STL10    | 86.8    | 73.0  | 49.8 | 0.89           | 0.92           | 1.04e-2  |
> > | Independent    | SVHN     | 57.5    | 50.6  | 80.5 | 0.12           | 0.0001         | 2.96e-1  |
> > | Independent    | CIFAR100 | 73.8    | 61.7  | 67.7 | 0.63           | 0.001          | 3.67e-1  |
> > | Independent    | STL10    | 79.4    | 73.1  | 55.8 | 0.34           | 0.001          | 5.21e-1  |

---

> ### Author Response · Authors · 2022-08-07
> **Have the concerns been addressed?**
>
> We would like to follow up to check whether the reviewer's concerns have been addressed.
>
> Based on the reviewer's suggestions we have:
> 1. motivated better the additional scores and clarified their correspondence to fidelity;
> 2. updated and added results to Table 1;
> 3. provided evaluation for MSE and SoftNN loss functions;
> 4. tried other density estimation models such as normalizing flows but found them inadequate for our task;
> 5. added the section on limitations to the (revised) main paper with the suggested modifications;
> 6. proposed an adaptive attack against our defense based on random obfuscations.

---

> > ### Comment · Reviewer_Ay7D · 2022-08-08
> > **Response to Rebuttal**
> >
> > I thank the authors for their thorough response to my concerns, and for clarifying the misunderstandings. I especially appreciate the explicit acknowledgement of limitations, and the updated results.
> >
> > Here are some follow-up comments:
> > - Is Appendix A needed? It seems to me that it has the same content as Section 5.4.
> > - Appendix E states *We observe that the smaller number of components for GMMs, the better the defense is*. Shouldn't this statement be flipped? It seems that increasing components makes the defense better.
> > - The updated results on CIFAR10 with 20 components look better, please update the statements on L306 and L652 accordingly.
> > - The new results however raise the importance of Appendix E and the discussion on the number of components. This appendix should ideally be summarized in the paper, and at worst referenced when discussing the main results. It would also be beneficial to substantiate the drawbacks of increasing components by (i) measuring training runtime (ii) demosntrating the case of overfitting that is stated.
> >
> > I am positive about the paper and believe it is a worthy contribution to the literature, thus I raise my score. The authors solve a relevant problem in a sound way (and are the first to do so), and have in my opinion adequately responded to all major concerns of reviewers. Nonetheless, I am looking forward to reading responses of other reviewers and find it necessary that they acknowledge that their concerns were appropriately covered, or continue the discussion otherwise.

---

> > > ### Author Response · Authors · 2022-08-08
> > > **Removed Appendix A, clarified and added statements on the number of components for GMMs.**
> > >
> > > We thank the reviewer for the response and further comments, we really appreciate the positive feedback.
> > >
> > > These are our new modifications:
> > > 1. We removed Appendix A (since it was the same content as in the new Section 5.4).
> > > 2. We resolved the inconsistencies and revised the paper.
> > > 3. We added the correct statement regarding the number of components for GMMs to the main paper in Section 5.2 and referenced the Appendix: *"We observe that the larger the number of components for GMMs, the better the defense is. We describe our approach to the selection of components for GMMs in Appendix D."* (Now it is Appendix D instead of E since we removed Appendix A on the limitations of our work).
> > >
> > > We will add the runtime for training GMMs and elaborate on the case of overfitting.

---

> > > > ### Author Response · Authors · 2022-08-09
> > > > **GMMs: runtime for training and overfitting**
> > > >
> > > > We updated Table 6 and Section D in Appendix. We show that, for example, for $150$ components for GMM on CIFAR10, the training time is 962.07 seconds, which is around $23$X longer than for $20$ components. Moreover, GMM overfits the training set for $150$ components since the p-values for encoders $f_s$ stolen with SVHN and STL10 queries are relatively high (> 0.05), while the p-value for the victim encoder $f_v$ is relatively low ($8.42e-10$).

---

### Official Review · Reviewer_z6xt · 2022-07-09

**Rating:** 5
**Confidence:** 4
**Soundness:** 4 excellent
**Presentation:** 3 good
**Contribution:** 3 good

**Summary:**

This paper generalizes the dataset inference defend originally proposed for supervised learning to self-supervised learning. Focusing on the model stealing black-box attack,  the authors utilize GMM to capture the density discrepancy between training and testing datasets of an adversary model. Furthermore, the authors propose new evaluation metrics based on mutual information and cosine similarity, and demonstrate the effectiveness and robustness through extensive experiments. Overall, it is an interesting paper.

**Questions:**

- See the Weakness part.

**Limitations:**

- See the Weakness part.

**Strengths And Weaknesses:**

- Strengths:
  - The motivation is clearly explained in the introduction to demonstrate that adaptation of data inference defend for SSL models is non-trivial.
  - Extensive experiments are provided to show the effectiveness of the proposed defending towards model stealing attack.
  - The code is provided in the supplementary. I appreciate the authors to do that.

- Weakness:

  - The proposed method heavily relies on the assumption that the stolen model shares similar distribution in the feature space on the private training dataset with the victim model. If so, I'm curious whether the proposed method can only work when the stolen model is well-trained, which is to some extent proved by Table 3 (e.g., in the bottom part, when the number of queries is 5K, the p-values is 1.23e-1 > 0.05).
  - I think a common problem for density estimation related algorithms to work in reality is the generality of the confidence threshold. In your case, this is the p-value threshold which has been set as 0.05 according to line 193. Although it works in your experiments, we can see in Table 3, the range of p-values might vary a lot under different datasets.
  - I wonder how the robustness of dataset inference to obfuscations comes, especially for shuffle, which will totally disrupt the high-dimensional space and GMM. Could you provide further explanation about this property?

  - It would be stronger to show the effectiveness of the proposed mutual information score and cosine similarity score as evaluation metrics for defending if they are applicable for different defending methods instead of only the proposed data inference defending, like the watermark defense mentioned in line 95-103.

  - Writing:
    - In Section 2 (e.g., line 95-103), it seems that the terminology "encoder" has been viewed the same with "self-supervision pre-trained encoder" by default, which is not the case.
    - It would be better to put the theorem in the main paper with the proof remained in the appendix, like Theorem 1.1 in line 266.

---

> ### Author Response · Authors · 2022-08-02
> **Q7: It would be better to put the theorem in the main paper with the proof remained in the appendix, like Theorem 1 in line 266.**
>
> Thank you for the suggestion. We moved Theorem 1 to the main part of the paper and have kept the proof in the Appendix (I.1). Please, see the revised paper.

---

> ### Author Response · Authors · 2022-08-02
> **Q6: In Section 2 (e.g., line 95-103), it seems that the terminology "encoder" has been viewed the same with "self-supervision pre-trained encoder" by default, which is not the case.**
>
> We modified the writing to make this clearer. We use the same terminology as in the previous paper on this topic [14], where the term “encoder” is used in the context of self-supervised learning and denotes a pre-trained model using self-supervised methods such as contrastive learning. We used “SSL encoder” in the answers to avoid any ambiguity.
>
> **References:**
>
> [14] [On the difficulty of defending self-supervised learning against model extraction.](https://proceedings.mlr.press/v162/dziedzic22a.html) Adam Dziedzic, Nikita Dhawan, Muhammad Ahmad Kaleem, Jonas Guan, and Nicolas Papernot.  ICML, 2022.

---

> ### Author Response · Authors · 2022-08-02
> **Q5: It would be stronger to show the effectiveness of the proposed mutual information score and cosine similarity score as evaluation metrics for defending if they are applicable for different defending methods instead of only the proposed data inference defending, like the watermark defense mentioned in line 95-103.**
>
> The mutual information and cosine similarity scores are primarily used to measure the quality of stolen SSL encoders and not to assess the performance of defense methods.

---

> ### Author Response · Authors · 2022-08-02
> **Q4: I wonder how the robustness of dataset inference to obfuscations comes, especially for shuffle, which will totally disrupt the high-dimensional space and GMM. Could you provide further explanation about this property?**
>
> Our defense compares representations for train and test sets obtained from a single given SSL encoder (we do not compare representations made by different SSL encoders, as explained in the answer to [Q1](https://openreview.net/forum?id=CCBJf9xJo2X&noteId=g2JM5VrgrmJ)). The shuffling changes the order of elements in the representation vectors only between the victim and stolen encoders, however, the order of elements in the representation vectors remains the same for each answered query from a given SSL encoder. Then, GMMs are still able to either differentiate between the distributions of train and test representations or mark them as similar for a given SSL encoder. If we shuffled representations from a given SSL encoder for each query differently, then these representations would not be useful for legitimate users on their downstream tasks.

---

> ### Author Response · Authors · 2022-08-02
> **Q3: I think a common problem for density estimation related algorithms to work in reality is the generality of the confidence threshold. In your case, this is the p-value threshold which has been set as 0.05 according to line 193. Although it works in your experiments, we can see in Table 3, the range of p-values might vary a lot under different datasets.**
>
> The verifier can select the significance level (confidence threshold) according to their tests on the victim encoder (being stolen) and independent SSL encoders. Our empirical analysis shows that selecting the standard confidence threshold of 0.05 works as expected for our defense. The range of p-values might vary substantially, however, the main point is that the differences between them are sufficient to distinguish between stolen vs independent SSL encoders.

---

> ### Author Response · Authors · 2022-08-02
> **Q2: If so, I'm curious whether the proposed method can only work when the stolen model is well-trained, which is to some extent proved by Table 3 (e.g., in the bottom part, when the number of queries is 5K, the p-values is 1.23e-1 > 0.05).**
>
> In general, we show that the higher quality of the extracted encoder, the higher confidence of our defense in resolving the encoder as being stolen. Indeed, as shown empirically in Table 3. Specifically, our method marks an encoder as stolen if, for the standard significance level (confidence threshold) of 0.05, the SVHN encoder is queried with 10K queries, while the ImageNet encoder with 40K queries (we updated Table 3, please see the revised paper).

---

> ### Author Response · Authors · 2022-08-02
> **Q1: The proposed method heavily relies on the assumption that the stolen model shares a similar distribution in the feature space on the private training dataset with the victim model.**
>
> We do not compare representations from different SSL encoders in our dataset inference method. Instead, we compare the representations from a single SSL encoder via GMMs. To elaborate on our defense method, we create a separate GMM for each potentially stolen SSL encoder. Then, we pass the private train and test data samples through the SSL encoder to obtain the respective representations that are further passed through the GMM and compared in terms of their likelihood (current Figure 7 in the revised paper, previously it was Figure 1, which shows this process in detail). Additionally, the obfuscation methods demonstrate that even when the adversary modifies its representations, the fact that we compare the train and test representations only from a single SSL encoder, it enables us to detect the model theft. Thus, a similar distribution in the feature space between the victim and stolen SSL encoders is not required for our method to work.

---

> ### Author Response · Authors · 2022-08-02
> **Responses to Reviewer z6xt**
>
> We appreciate the positive, encouraging, and constructive feedback. We thank the reviewer for the detailed analysis of our paper and below provide a case-by-case response to the comments.

---

> ### Author Response · Authors · 2022-08-07
> **Have the concerns been addressed?**
>
> We would like to follow up to check whether the reviewer's concerns have been addressed.
>
> Based on the reviewer's suggestions we have:
> 1. elaborated on our defense method where we compare the representations from a single SSL encoder via GMMs;
> 2. showed that the higher quality of the extracted encoder, the higher confidence of our defense;
> 3. provided the empirical analysis to show that selecting the standard confidence threshold of 0.05 works as expected for our defense;
> 4. explained the robustness of dataset inference to obfuscations;
> 5. defined the mutual information and cosine similarity scores as metrics to measure the quality of stolen SSL encoders;
> 6. addressed the comments on terminology as well as the placement of Theorem 1 in the (revised) main paper;

---

> ### Comment · Reviewer_z6xt · 2022-08-09
> **Reply to the authors**
>
> I would like to thank the authors for addressing my concerns during the short rebuttal period. I will keep my rating as "Borderline accept".

---

> > ### Author Response · Authors · 2022-08-09
> > **Thank you for the review & positive feedback**
> >
> > We thank the reviewer for the discussion and appreciate the positive feedback. If there is anything more we could further improve, please let us know.

---

### Official Review · Reviewer_FsB3 · 2022-07-10

**Rating:** 7
**Confidence:** 4
**Soundness:** 3 good
**Presentation:** 4 excellent
**Contribution:** 3 good

**Summary:**

The authors propose a detection method for dataset inference for the unsupervised setting of models. Unlike prior work on models meant for classification, the authors in this work focus on models trained to generate feature representations (SSL). The detection technique relies on log-likelihood statistics using density estimation models, along with measures of distributions like MI. The proposed method has promising results, taking a step in the direction of making dataset inference broader and more useful in the real world.

**Questions:**

# Minor comments

- Line 34: Isn't 'dataset inference' the threat model? Like membership inference or property inference?
- Figure 1 is too complicated- either increase the size (and de-clutter) or simplify the flow/structure.
- Line 342: "implies" -> "suggests"
- Line 347: "defense" -> "detection". At the point in time where the method is launched, the adversary has reportedly already stolen the model, and thus it is merely the question of detecting (or confirming) whether that is true, not preventing the adversary from stealing the victim's model.
- Tables 2,3 seem to have overfill issues.

**Limitations:**

Addressed in Section 6

**Strengths And Weaknesses:**

# Strengths

- The paper is well written and sets up the threat model, as well as the proposed method(s) and the rationale behind them, quite well.
- Empirical evaluation is thorough, with datasets ranging from CIFAR to ImageNet.

# Weaknesses

- Lines 203-205. As a baseline, the authors should include experiments for scenarios where downstream tasks (single, or 2-3 combined) are trained using the "feature extractors". It's true that "a single downstream task cannot adequately reflect...", but it would be nice to have some estimates (performance numbers) to know for sure how much this difference really is.
- Lines 322-322: How about adding noise in intermediate layers (or inputs/outputs), or dropout?
- Table 2: Performance difference between $f_s$ for some cases does not seem to be too far from $f_i$

---

> ### Author Response · Authors · 2022-08-02
> **Noise in intermediate layers, Table 2, Threat Model, Figure 1, and Minor Comments**
>
> >**Lines 322-322: How about adding noise in intermediate layers (or inputs/outputs), or dropout?**
>
> Given the structure of neural networks, where intermediate layers affect subsequent ones and the errors accumulate, these types of noise additions would lead to a functionally different adversary model. While this would likely avoid the model being detected as stolen, it would also negatively affect the utility of the model for the adversary. As part of our defense setup, we assume that the encoder provided by the adversary to the arbitrator is the same as the encoder it uses normally. Otherwise, the adversary could provide an unrelated model to the arbitrator from where detecting whether a model is stolen would not be possible.
>
> There have been defenses based on a similar idea of random perturbation of intermediate layers in the area of adversarial examples [1]. However, such “noisy” networks have a drop in their performance and there are methods to recover obfuscated outputs, such as EOT (Expectation Over Transformation) [2]. In our case, the defender could query the noisy encoder many times with the same input and take the averaged representation so that the effect of added noise is minimized.
>
> **References:**
>
> [1] [Towards Robust Neural Networks via Random Self-ensemble.](https://openaccess.thecvf.com/content_ECCV_2018/html/Xuanqing_Liu_Towards_Robust_Neural_ECCV_2018_paper.html)
> Xuanqing Liu, Minhao Cheng, Huan Zhang, Cho-Jui Hsieh. ECCV 2018.
>
> [2] [Synthesizing Robust Adversarial Examples.](https://proceedings.mlr.press/v80/athalye18b.html) Anish Athalye, Logan Engstrom, Andrew Ilyas, Kevin Kwok. ICML 2018.
>
>
> >**Table 2: Performance difference between $f_s$ for some cases does not seem to be too far from $f_i$.**
>
> We discussed the details of scores and their performance tradeoffs in the main paper in Section 5.3. In general, we see that for the mutual information scores, the differences between $f_s$ and $f_i$ are not very large, which is likely because the mutual information score is based on an approximation. Even though the differences are small, they are sufficient to distinguish between the stolen and independent encoders in most cases. In contrast, the cosine similarity score has large differences between $f_s$ and $f_i$ when no obfuscations are applied but is not robust under obfuscations where the score for $f_s$ becomes almost equivalent to $f_i$.
>
> >**Line 34: Isn't *dataset inference* the threat model? Like membership inference or property inference?**
>
> Dataset inference is not a threat model. The original paper on dataset inference describes it as “the process of identifying whether a suspected model copy has private knowledge from the original model’s dataset, as a defense against model stealing” in the abstract and we use the same formulation in our work.
>
> >**Figure 1 is too complicated - either increase the size (and de-clutter) or simplify the flow/structure.**
>
> We simplified Figure 1 in the revised paper, where we present only the ownership resolution for a single potentially stolen SSL encoder. The full overview of our method (also slightly simplified), where we show each stage of the dataset inference, was moved to the new section J in the Appendix as Figure 7.
>
> >**Minor Comments.**
>
> The overfill issues for Tables 2 and 3 and the comment on line 342 have also been fixed. Regarding the comment for line 347, we would like to clarify that we consider our method as a *reactive* defense, which works after a model has been stolen; however for consistency with [15,30] we refer to it as a defense.
>
> **References:**
>
> [15] [Increasing the cost of model extraction with calibrated proof of work](https://openreview.net/forum?id=EAy7C1cgE1L). Adam Dziedzic, Muhammad Ahmad Kaleem, Yu Shen Lu, and Nicolas Papernot. ICLR 2022.
>
> [30] [EncoderMI: Membership Inference against Pre-trained Encoders in Contrastive Learning](https://dl.acm.org/doi/10.1145/3460120.3484749). Hongbin Liu, Jinyuan Jia, Wenjie Qu, Neil Zhenqiang Gong. CCS 2021.

---

> ### Author Response · Authors · 2022-08-02
> **Downstream tasks and the new metrics (mutual information and cosine similarity) to measure the quality of stolen encoders**
>
> Thank you for your detailed, helpful, and insightful feedback. We address individual points below.
>
> >**Lines 203-205. As a baseline, the authors should include experiments for scenarios where downstream tasks (single, or 2-3 combined) are trained using the "feature extractors". It's true that "a single downstream task cannot adequately reflect...", but it would be nice to have some estimates (performance numbers) to know for sure how much this difference really is.**
>
> We have already included these numbers in the Appendix H and Tables 7, 8, and 9. We have also modified Table 9 and combined it with results from Tables 1, 2, and 3, to incorporate the performance on downstream tasks and our new metrics, namely the mutual information score, cosine similarity, and the p-values. Table 9 is for the victim encoder pre-trained on the SVHN dataset. We also added a corresponding Table 10 for the victim encoder that is pre-trained on the CIFAR10 dataset. Please, see our revised paper and the appendix in the supplementary materials.
>
> We present below our new results from Table 9. We observe that an encoder of high quality is extracted after around 7K queries according to the downstream tasks and our metrics show evidence of stealing after around 9K queries. Thus, the linear evaluation on downstream tasks and our metrics indicate model stealing after a similar number of queries used by an adversary. We also note that the cosine similarity score has an almost linear dependence on the number of queries that the adversary used to extract the victim encoder, which is helpful to adequately measure the quality of this stolen encoder.
>
> | \# of Queries  | Dataset  | CIFAR10 | STL10 | SVHN | $S(\cdot,f_v)$ | $C(\cdot,f_v)$ | p-value  |
> |----------------|----------|---------|-------|------|----------------|----------------|----------|
> | Victim Encoder | N/A  	| 57.5	| 50.6  | 80.5 | 1          	| 1          	| 9.69e-227 |
> | 500        	| CIFAR10	| 24.5	| 23.3  | 19.3 | 0.00       	| 0.24       	| 6.89e-1  |
> | 5K         	| CIFAR10	| 53.3	| 45.2  | 58.1 | 0.11       	| 0.40       	| 3.51e-1  |
> | 7K         	| CIFAR10	| 57.9	| 50.9  | 69.2 | 0.14       	| 0.46       	| 4.72e-1  |
> | 8K         	| CIFAR10	| 59.6	| 52.0  | 72.6 | 0.53       	| 0.47       	| 9.87e-2  |
> | 9K         	| CIFAR10	| 59.9	| 50.8  | 71.5 | 0.57       	| 0.49       	| 6.23e-2  |
> | 10K        	| CIFAR10	| 59.1	| 51.3  | 72.3 | 0.69       	| 0.52       	| 5.82e-3  |
> | 20K        	| CIFAR10	| 60.6	| 51.5  | 73.4 | 0.92       	| 0.58       	| 2.31e-7  |
> | 30K        	| CIFAR10	| 60.7	| 52.1  | 74.1 | 0.93       	| 0.63       	| 2.11e-10  |
> | 50K        	| CIFAR10	| 59.8	| 51.6  | 75.1 | 0.94       	| 0.69       	| 1.19e-17  |
> | 50K        	| STL10           | 60.6	| 51.5  | 76.8 | 0.95       	| 0.89       	| 1.65e-11  |
> | 50K        	| SVHN	| 55,6	| 48.7  | 82.2 | 0.96       	| 0.91       	| 1.05e-75 |
> | Independent	| CIFAR10	| 87.4	| 73.4  | 49.5   | 0.90       	| 0.009     	| 3.56e-1  |
> | Independent	| CIFAR100 | 73.8	| 61.7  | 67.7 | 0.90       	| 0.007      	| 2.13e-1  |
> | Independent	| STL10	| 79.4	| 73.1  | 55.8 | 0.84       	| 0.015      	| 4.62e-1  |

---

> > ### Author Response · Authors · 2022-08-02
> > **New results from Table 10 for the comparison between the accuracy on downstream tasks vs new metrics**
> >
> > Next, we also present our new results from Table 10. For example, we observe that when stealing with SVHN queries from the CIFAR10 victim encoder, we cannot identify the stolen encoder as extracted from the victim based on the victim's performance on downstream tasks. On the other hand, our metrics show evidence of stealing after around 30K queries. Additionally, our metrics correctly demonstrate a negligible similarity between the victim and independent encoders.
> >
> > | \# of Queries  | Dataset  | CIFAR10 | STL10 | SVHN | $S(\cdot,f_v)$ | $C(\cdot,f_v)$ | p-value  |
> > |----------------|----------|---------|-------|------|----------------|----------------|----------|
> > | Victim Encoder | N/A      | 87.4    | 73.4  | 49.5 | 1              | 1              | 1.37e-16 |
> > | 500            | SVHN     | 22.3    | 20.8  | 20.1 | 0.00           | 0.07           | 5.69e-1  |
> > | 1K             | SVHN     | 26.5    | 23.4  | 29.9 | 0.06           | 0.11           | 9.02e-1  |
> > | 2K             | SVHN     | 40.0    | 35.2  | 46.7 | 0.12           | 0.13           | 7.83e-1  |
> > | 3K             | SVHN     | 44.4    | 39.8  | 56.3 | 0.18           | 0.21           | 8.04e-1  |
> > | 4K             | SVHN     | 47.4    | 42.5  | 60.8 | 0.22           | 0.19           | 6.30e-1  |
> > | 5K             | SVHN     | 49.5    | 43.8  | 60.9 | 0.29           | 0.28           | 4.28e-1  |
> > | 10K            | SVHN     | 55.6    | 46.6  | 58.9 | 0.38           | 0.36           | 6.42e-1  |
> > | 20K            | SVHN     | 57.7    | 48.7  | 60.4 | 0.43           | 0.39           | 9.81e-2  |
> > | 30K            | SVHN     | 58.0    | 49.2  | 57.0 | 0.57           | 0.45           | 6.73e-2  |
> > | 40K            | SVHN     | 61.4    | 51.1  | 55.0 | 0.69           | 0.49           | 4.83e-2  |
> > | 50K            | SVHN     | 61.2    | 51.7  | 54.8 | 0.76           | 0.50           | 3.97e-2  |
> > | 50K            | CIFAR10  | 84.8    | 70.7  | 52.4 | 0.84           | 0.95           | 8.73e-7  |
> > | 50K            | STL10    | 86.8    | 73.0  | 49.8 | 0.89           | 0.92           | 1.04e-2  |
> > | Independent    | SVHN     | 57.5    | 50.6  | 80.5 | 0.12           | 0.0001         | 2.96e-1  |
> > | Independent    | CIFAR100 | 73.8    | 61.7  | 67.7 | 0.63           | 0.001          | 3.67e-1  |
> > | Independent    | STL10    | 79.4    | 73.1  | 55.8 | 0.34           | 0.001          | 5.21e-1  |

---

> ### Author Response · Authors · 2022-08-07
> **Have your concerns been addressed?**
>
> We would like to follow up to check whether the reviewer's concerns have been addressed.
>
> Based on the reviewer's suggestions we have:
> 1. combined the linear evaluation on downstream tasks and our metrics into a single table and ran additional experiments to present differences between the metrics;
> 2. elaborated on adding noise or dropout as a form of attack on our defense;
> 3. explained more the performance difference between stolen encoder $f_s$ and an independent one $f_i$;
> 4. defined clearly the dataset inference as "the process of identifying whether a suspected model copy has private knowledge from the original model’s dataset, as a defense against model stealing";
> 5. updated Figure 1 to simplify its flow/structure;
> 6. addressed minor comments.

---

> > ### Comment · Reviewer_FsB3 · 2022-08-08
> > **Issues Addressed**
> >
> > Hello authors,
> >
> > Thanks for the detailed reubuttal. Yes, most of my issues have been addressed, and I am now adjusting my scores to reflect that. Best of luck!

---

> > > ### Author Response · Authors · 2022-08-08
> > > **Thank you for the review & increasing the score**
> > >
> > > We thank the reviewer for the discussion and appreciate the positive feedback.

---

### Official Review · Reviewer_g8VY · 2022-07-12

**Rating:** 5
**Confidence:** 3
**Soundness:** 2 fair
**Presentation:** 3 good
**Contribution:** 3 good

**Summary:**

This paper investigates dataset inference for encoders. Dataset inference is a defense against model stealing, which allows the defender to identify whether a suspected stolen model has private knowledge from the original model’s training data. Originally formulated for classifiers, the authors adapt dataset inference to encoders by testing for differences in encoder representations on the defender’s training data vs. held-out data. Specifically, they fit a GMM to estimate the distribution of the suspected stolen encoder’s representations on training data. They then use the GMM to evaluate the likelihood of the encoder’s representations: if the encoder is stolen then the likelihood of the training data is expected to be higher than the likelihood of the held-out data. Experiments in the vision domain demonstrate that the method successfully identifiers stolen encoders with high confidence. The confidence is shown to improve when the stolen encoder’s representations are closer to the original, as measured by mutual information and cosine similarity.

**Questions:**

Is it possible to make this approach more practical given the issues raised above?

Can this approach handle more subtle forms of stealing? E.g., where an encoder is initialized by model extraction, then refined on independent training data. Such cases are apparently handled in the original formulation of data inference by Maini et al. (2020).

It would be good to elaborate on what makes GMMs suited for this task. As I understand, GMMs are not universal approximators when the number of components is fixed. In practice, how would one choose the number of components? Could other density estimators be used?

**Limitations:**

Limitations are discussed in appendices, however a summary should be included in the main paper.

**Strengths And Weaknesses:**

*Quality*

The idea of comparing encoding representations on private training data vs held-out data makes sense. However some aspects of the approach are not practical:
- The defender is required to reveal private training/test data to a trusted arbitrator. This is not required in the original formulation for classifiers (Maini et al., 2021), where the defender is able to run dataset inference themselves given query access to the stolen model.
- In order to run dataset inference, the defender’s private train/test data sets must be passed through the adversary’s encoder (in their entirety). This is problematic if it is necessary to query the adversary’s encoder via a public API under their control. The adversary could simply steal the defender’s private data. One way around this would be to compel the adversary to share their model with the trusted arbitrator. However this may not be possible from a legal perspective. Note that this is not an issue for the original formulation by Maini et al. (2021), where it is only necessary to pass a small fraction of the private training set through the adversary’s model.

*Originality and Significance*

While dataset inference is not a new idea, it has only been applied to classifiers to date. Extensions to other models with more complex outputs, such as encoders, are well-motivated. There is some prior work on membership inference for encoders (Liu et al., 2021), which ought to be discussed in the paper. Given the close connection between membership inference and dataset inference, Liu et al.’s approach could be an alternative to the approach proposed in this paper. Another related work that ought to be cited is by (Jia et al., 2021). They discuss an alternative reactive defense for model stealing called proof-of-learning.

*Clarity*

The paper is clear overall. I wonder if Figure 1 could be simplified, as it seems many of the quantities do not need to be computed to run the test against a single potentially stolen encoder.

- Maini, Pratyush, Mohammad Yaghini, and Nicolas Papernot. "Dataset Inference: Ownership Resolution in Machine Learning." International Conference on Learning Representations. 2020.
- Liu, Hongbin, et al. "EncoderMI: Membership inference against pre-trained encoders in contrastive learning." Proceedings of the 2021 ACM SIGSAC Conference on Computer and Communications Security. 2021.
- Jia, Hengrui, et al. "Proof-of-learning: Definitions and practice." 2021 IEEE Symposium on Security and Privacy (SP). IEEE, 2021.

---

> ### Author Response · Authors · 2022-08-02
> **Proof-of-Learning, Limitations, and Figure 1**
>
> >**Another related work that ought to be cited is by (Jia et al., 2021). They discuss an alternative reactive defense for model stealing called proof-of-learning.**
>
> We added this paragraph in Section 2 on Related Work: *“Proof-of-Learning (Jia et al., 2021) is a reactive defense that involves the defender claiming ownership of a model by showing incremental updates of the model training. It is a complementary method to dataset inference, which instead identifies a stolen model. PoL could be applied directly to SSL encoders, however, it requires an expensive verification process, where the verifier needs to perform model updates and the prover needs to save intermediate weights of the model, which is more expensive than dataset inference with GMMs.”* Please, see the revised paper.
>
> We note that our defense and Proof-of-Learning are not alternatives of each other, but rather complementary: our defense helps identify an adversary's model as stolen from the owner's, whereas Proof-of-Learning helps verify that a model is trained by the owner.
>
> >**Limitations are discussed in appendices; however, a summary should be included in the main paper.**
>
> We added Section 5.4 as a summary of Limitations to the main paper: *“If the t-test run as part of dataset inference is inconclusive for an extracted encoder, we cannot state whether the encoder was stolen. Similarly, for an independent encoder, there is the possibility of it being incorrectly classified as stolen. Previous work [26,38] has shown that self-supervised encoders trained using heavy augmentations and contrastive learning generalize better than their supervised counterparts, which makes it harder for the dataset inference to differentiate between train and test representations in SSL than in the SL setting [14]. The loss values of projected individual representations are insufficient for dataset inference [14]. We build on top of this observation to enable dataset inference for encoders and use GMMs to distinguish between train and test representations.”*
>
> >**I wonder if Figure 1 could be simplified, as it seems many of the quantities do not need to be computed to run the test against a single potentially stolen encoder.**
>
> We simplified Figure 1 in the revised paper, where we present only the ownership resolution for a single potentially stolen SSL encoder. The full overview of our method (also slightly simplified), where we show each stage of the dataset inference, was moved to section J in the Appendix as Figure 7.

---

> ### Author Response · Authors · 2022-08-02
> **Refining stolen encoders and suitability of GMMs**
>
> >**Can this approach handle more subtle forms of stealing? E.g., where an encoder is initialized by model extraction, then refined on independent training data.**
>
> We thank the reviewer for this suggestion. We ran additional experiments to evaluate this setting and observed that our approach is able to identify fine-tuned models as stolen. We use a stolen encoder from the SVHN victim model (stolen with SVHN data) and then retrain it with standard contrastive learning using 5 epochs on the following number of samples (5K, 10K, 25K, 50K) from CIFAR10. We then check how our dataset inference performs on the fine-tuned encoders. As expected, the p-values increase with more samples for fine-tuning, while the values of the effect size decrease.
>
> | # of samples | p-value | $\Delta \mu$ (effect size) |
> |---------|---------------|--------|
> | 5K        |  3.24e-16    | 7.03
> | 10K        |  1.82e-16    | 7.14     |
> | 20K        | 6.91e-14     | 6.09     |
> | 50K        | 5.28e-12     | 5.53     |
>
> We also check how the number of epochs over which the fine-tuning is done influences our defense. We use the same stolen encoder as in the previous experiment and keep 50K as the number of data points used for fine-tuning. We also observe that with more epochs, p-values increase while the values of the effect size decrease.
>
> | # of epochs | p-value | $\Delta \mu$ (effect size) |
> |---------|---------------|--------|
> | 5        |  5.28e-12    |5.53     |
> | 10        | 8.73e-6     |4.62     |
> | 25        | 6.81e-1     |1.34     |
> | 50        | 1.73e-1     |0.92     |
> | 100      | 8.53e-1     | -0.53     |
>
> We note that in the original work on dataset inference for supervised models [31], the fine-tuning was done directly on the victim model (which was directly accessed by an adversary) instead of on a stolen model. Additionally, the extra computation required by the adversary for fine-tuning diminishes the benefit of stealing the encoder.
>
> >**It would be good to elaborate on what makes GMMs suited for this task. As I understand, GMMs are not universal approximators when the number of components is fixed. In practice, how would one choose the number of components?**
>
> We chose GMMs because other density estimation models like normalizing flows are overparameterized and would require more training data. In essence, we do not need to model the density of the representations very accurately: our only requirement is for the method to distinguish between the densities of the representations.
>
> Given an appropriate number of components, the use of GMMs is justified by the Central Limit Theorem. They enjoy theoretical properties of being consistent and asymptotically normal estimators. The efficiency of training a GMM trades off its simplicity relative to other more complex parametric density estimation methods.
>
> A Gaussian mixture model is a universal approximator of densities, in the sense that any smooth density can be approximated with any speciﬁc nonzero amount of error by a Gaussian mixture model with enough components [1]. In our method, we do not fix the number of components but find the number of components so that we are able to (1) correctly mark the victim model as the original model trained on the given private training set and (2) obtain a non-conclusive answer (indicating that we cannot state that the model was stolen) for an independent model. Then, a GMM with this number of components is used to assess the potentially stolen SSL encoder.
>
> There are several proposed methods in the literature to choose the number of components for a GMM, based on validation, mutual information, or AIC and BIC scores [2] (more details can be found [here](https://citeseerx.ist.psu.edu/viewdoc/download?doi=10.1.1.109.8192&rep=rep1&type=pdf)). However, we observe that our setting allows us to efficiently find an appropriate number of components.
>
> **References:**
>
> [1] [Deep Learning (book)](https://www.deeplearningbook.org/). Ian Goodfellow, Yoshua Bengio, and Aaron Courville. 2016.
>
> [2] [The estimating optimal number of Gaussian mixtures based on incremental k-means for speaker identification](https://citeseerx.ist.psu.edu/viewdoc/download?doi=10.1.1.109.8192&rep=rep1&type=pdf). Younjeong Lee, Ki Yong Lee, Joohun Lee. International Journal of Information Technology, 2006.
>
> [31] [Dataset Inference: Ownership Resolution in Machine Learning](https://openreview.net/forum?id=hvdKKV2yt7T). Pratyush Maini, Mohammad Yaghini, Nicolas Papernot. ICLR 2021.

---

> ### Author Response · Authors · 2022-08-02
> **Trusted arbitrator and on how to make the method more practical**
>
> >**One way around this would be to compel the adversary to share their model with the trusted arbitrator. However this may not be possible from a legal perspective.**
>
> This is indeed how we envision an application of dataset inference in the practical setting. As the defender shares both a model and their private data with an arbitrator, we work under the assumption that it is not unreasonable for the arbitrator to also request the adversary’s model. This assumption has precedent in this field; for example, proof-of-learning, which the reviewer aptly points out as another related work, requires a third-party verifier with access to model weights acting as arbitrator ("the verifier would be a legal entity resolving ownership disputes" [1]).
>
> Another alternative is to use protocols from cryptography such as homomorphic encryption and secure multi-party computation for private inference (inference on encrypted data), which would avoid the need for the victim to reveal the private data points [2,3].
>
> >**Is it possible to make this approach more practical given the issues raised above?**
>
> To summarize the responses above, we believe two changes can address the issues raised:
>
> 1. The defender can reveal a smaller subset of their training data at the expense of lower confidence in the outcome of dataset inference.
> 2. Concerns around the direct querying of the adversary’s model with private training data from the defender can be addressed using cryptographic primitives such as homomorphic encryption and/or a secure multiparty computation approach for private inference [2,3].
>
> **References:**
>
> [1] [Proof-of-learning: Definitions and practice.](https://arxiv.org/abs/2103.05633) Hengrui Jia, Mohammad Yaghini, Christopher A. Choquette-Choo, Natalie Dullerud, Anvith Thudi, Varun Chandrasekaran, Nicolas Papernot. IEEE Symposium on Security and Privacy (S&P), 2021.
>
> [2] [MP2ML: A Mixed-Protocol Machine Learning Framework for Private Inference.](https://encrypto.de/papers/BCDSY20.pdf) Boemer, Fabian and Cammarota, Rosario and Demmler, Daniel and Schneider, Thomas and Yalame, Hossein. 2020.
>
> [3] [Delphi: A Cryptographic Inference Service for Neural Networks.](https://www.usenix.org/conference/usenixsecurity20/presentation/mishra) Pratyush Mishra and Ryan Lehmkuhl and Akshayaram Srinivasan and Wenting Zheng and Raluca Ada Popa. USENIX, 2020.

---

> ### Author Response · Authors · 2022-08-02
> **Number of queries for dataset inference**
>
> >**In order to run dataset inference, the defender’s private train/test data sets must be passed through the adversary’s encoder (in their entirety).**
>
> In our method, we do not require to pass the datasets in their entirety through the adversary’s encoder. Our method does require a significantly larger number of private samples compared to the original formulation of dataset inference. However, this is primarily because of the fundamental difference between the supervised and self-supervised settings and the size of the output representations where getting a signal that is statistically significant requires a larger number of queries.
> We run additional experiments, where we compute the p-values for ownership resolution with different numbers of private data points used and show that the signal becomes weaker as the number of utilized data points decreases.
>
> In the Table below (Table 17 in the Appendix), we detect if a given encoder was stolen using a different number of queries. $D_P$ is the victim's private data, $D_S$ is the data used for stealing. *# GMM* is the number of data points used to train the GMM, *# train* is the number of private data points and # test is the number of test data points that are used to compare (in terms of the representations) using GMM. *# total* is the total number of data points used for our dataset inference method. We observe that for the ImageNet victim stolen with SVHN queries, our method requires at least 30K queries for GMMs and 10K for both private training and test data points. For the SVHN victim stolen with SVHN queries, our method requires at least 5K queries for GMMs and 1K for both private training and test data points.
>
>
> | $D_P$    | $D_S$ | \# GMM | \# train | \# test | \# total | p-value  | $\Delta \mu$ (effect size) |
> |----------|-------|--------|----------|---------|----------|----------|--------------|
> | ImageNet | SVHN  | 100K   | 50K      | 50K     | 200K     | 3.33e-4  | 14.79        |
> | ImageNet | SVHN  | 50K    | 50K      | 50K     | 150K     | 1.11e-3  | 10.76        |
> | ImageNet | SVHN  | 50K    | 25K      | 25K     | 100K     | 8.99e-3  | 9.68         |
> | ImageNet | SVHN  | 50K    | 10K      | 10K     | 70K      | 3.42e-2  | 8.26         |
> | ImageNet | SVHN  | 50K    | 5K       | 5K      | 60K      | 7.81e-2  | 6.32         |
> | ImageNet | SVHN  | 40K    | 10K      | 10K     | 60K      | 2.81e-2  | 8.63         |
> | ImageNet | SVHN  | 40K    | 5K       | 5K      | 50K      | 9.72e-2  | 6.01         |
> | ImageNet | SVHN  | 30K    | 10K      | 10K     | 50K      | 4.52e-2  | 8.47         |
> | ImageNet | SVHN  | 30K    | 5K       | 5K      | 40K      | 1.23e-1  | 5.89         |
> | ImageNet | SVHN  | 20K    | 10K      | 10K     | 40K      | 6.52e-2  | 7.81         |
> | ImageNet | SVHN  | 10K    | 10K      | 10K     | 30K      | 4.52e-1  | 3.59         |
> | SVHN     | SVHN  | 50K    | 10K      | 10K     | 70K      | 8.23e-68 | 10.69        |
> | SVHN     | SVHN  | 50K    | 1K       | 1K      | 52K      | 7.63e-32 | 7.53         |
> | SVHN     | SVHN  | 10K    | 10K      | 10K     | 30K      | 9.80e-10 | 4.05         |
> | SVHN     | SVHN  | 10K    | 1K       | 1K      | 12K      | 2.36e-6  | 3.41         |
> | SVHN     | SVHN  | 5K     | 1K       | 1K      | 7K       | 7.33e-4  | 2.28         |
> | SVHN     | SVHN  | 1K     | 1K       | 1K      | 3K       | 4.56e-1  | 1.26         |

---

> > ### Comment · Reviewer_g8VY · 2022-08-10
> > **RE: Number of queries for dataset inference**
> >
> > Thanks for the response - it's great to see experiments varying the amount of private data revealed to the arbitrator. However, I don't see how this experiment is compatible with the description of the method given in Section 3.3 of the paper.
> >
> > On line 174, it is stated that the "victim's encoder $f_v$ is trained using the whole private training dataset $D_P$". On lines 183-184 the arbitrator is said to train a density estimator based on the representations of $D_{P2}$ from the stolen encoder $f_s$. Then in line 186, the arbitrator is said to generate representations of $D_{P1}$ and $D_N$ using the stolen encoder $f_s$.
> >
> > Thus it seems $D_P = D_{P1} \cup D_{P2}$ (the data used to train the victim's encoder) and $D_N$ (the test set) are all passed through the stolen encoder $f_s$.

---

> > > ### Author Response · Authors · 2022-08-10
> > > **Private Training Dataset**
> > >
> > > Thank you for the response. Unfortunately we did not update Section 3.3 according to the new experiments which is why the description there suggests that the whole private training dataset is used. In the new results above, $D_{P1}$ is \# GMM, $D_{P2}$ is \# train so that $D_P \neq D_{P1} \cup D_{P2}$. Similarly $D_N$ corresponds to \# test. We will update Section 3.3 to make this point clear.

---

> ### Author Response · Authors · 2022-08-02
> **Direct query access to a stolen encoder**
>
> We appreciate the constructive feedback. We thank the reviewer for the detailed analysis of our paper and provide a case-by-case response to the comments below.
>
> >**The defender is required to reveal private training/test data to a trusted arbitrator. This is not required in the original formulation for classifiers (Maini et al., 2021), where the defender is able to run dataset inference themselves given query access to the stolen model.**
>
> We strictly improve upon the original dataset inference, where the private data points are directly revealed to the potential adversary. Our method is compatible with direct query access, thus the defender can also send the private training data points to the potentially stolen encoder and obtain the corresponding representations to calculate the relevant log-likelihoods needed to train a density estimator for dataset inference. However, the adversary might capture our private data points as a result of this querying and claim that they also trained their encoder on these points. The arbitrator, as the 3rd trusted party, eschews this problem. While the presence of a trusted arbitrator is indeed an additional requirement in our method, this arrangement has been made primarily to allow for an increased level of transparency between the defender and a potential adversary in the process of dataset inference. We note that it is still possible for the defender to run dataset inference themselves.

---

> > ### Comment · Reviewer_g8VY · 2022-08-10
> > **RE: Direct query access to a stolen encoder**
> >
> > Thanks - this is a convincing explanation. I'd personally like to see a remark in the paper explaining why the formulation differs from Maini et al. (2021). Although I realize space is limited.

---

> ### Author Response · Authors · 2022-08-07
> **Have your concerns been addressed?**
>
> We would like to follow up to check whether the reviewer's concerns have been addressed.
>
> Based on the reviewer's suggestions we have:
> 1. showed that the dataset inference for encoders is compatible with the direct query access;
> 2. ran a new experiment to present the minimum number of queries required for our dataset inference, and in general, the dependence between the number of queries vs the confidence of our test;
> 3. added new results that show the impact of fine-tuning on stolen encoders;
> 4. elaborated more on the suitability of GMMs for our task;
> 5. cited suggested related work;
> 6. added a section on limitations to the revised paper;
> 7. updated Figure 1.

---

> ### Author Response · Authors · 2022-08-09
> **Pending questions**
>
> We would like to thank the reviewer for the questions and comments. The paper has definitely improved as a result. We would like to check one last time if there are any pending questions that we have not adequately addressed.

---

> > ### Comment · Reviewer_g8VY · 2022-08-10
> > **RE: Pending questions**
> >
> > Thanks for the thorough response.
> >
> > I'm satisfied with the author's justification for introducing a trusted arbitrator, especially since it has precedent in the model stealing literature.
> >
> > I'm still unsure about the issue of revealing the private dataset $D_P$ (used to train the encoder) to the arbitrator. The method as described in the paper seems to require that representations are produced for all instances in $D_{P}$ using the stolen encoder (see [previous comment](https://openreview.net/forum?id=CCBJf9xJo2X&noteId=i_fUcMMJ88W)). If this is not the case, then some refactoring may be needed to avoid confusion.
> >
> > I will consider revising my score during the reviewer-AC discussion period.

---

### Author Response · Authors · 2022-08-02
**General Response**

We thank the reviewers for their comments and feedback. We answer all the questions below in-line and provide a revised version of our paper (both the main and the appendix), where the new or updated parts are marked in blue.

---

### Meta-Review · Area_Chair_pH8J · 2022-08-26

**Recommendation:** Accept
**Confidence:** Less certain

**Metareview:**

This paper joins an interesting area that tackles the use of models in the real world that are accessible publicly via APIs. In these cases, there may be adversaries that  attempt to steal the model. This can be done by accessing information about the model from particular queries. One of the approaches used to tackle such scenarios is to develop defenses that can detect when such models are being stolen. Much of this area is focused on looking at supervised models; the authors introduce a similar approach for encoders.

In the encoder case, techniques that involve the decision boundary, suitable for supervised models, no longer apply. The authors provide some technical innovations to get around this issue. Essentially they look for evidence of memorization by using a metamodel.

Overall, the problem is well-motivated and the tools the authors develop are interesting; the results also appear convincing. The reviewers reached a near-consensus that the paper is worth accepting, and I agree. Nothing here is extremely groundbreaking, but it's a well-crafted approach to handle a case of an important problem that hasn't been addressed yet.

The authors generally responded to all of the questions the reviewers had, and I furthermore found the answers convincing. Ultimately I think it's worth accepting.

**Award:**

No

---

### Decision · Program_Chairs · 2022-09-14

Accept